# Nano-Selenium Modulates NF-κB/NLRP3 Pathway and Mitochondrial Dynamics to Attenuate Microplastic-Induced Liver Injury

**DOI:** 10.3390/nu16223878

**Published:** 2024-11-14

**Authors:** Qi Shen, Yunjie Liu, Jiakui Li, Donghai Zhou

**Affiliations:** College of Veterinary Medicine, Huazhong Agricultural University, Wuhan 430070, China; sysq@webmail.hzau.edu.cn (Q.S.);

**Keywords:** microplastics, nano-selenium, inflammation, mitochondrial dynamics, NF-κB/NLRP3 signaling pathway

## Abstract

Background: Microplastics (PS-MPs) are a new type of pollutant with definite hepatotoxicity. Selenium, on the other hand, has natural, protective effects on the liver. Objectives/Methods: The purpose of this experiment is to find out whether nano-selenium (SeNP) can alleviate liver damage caused by microplastics. Initially, we established through in vitro experiments that SeNP has the ability to enhance the growth of healthy mouse liver cells, while microplastics exhibit a harmful impact on normal mouse hepatocyte cell suspensions, leading to a decrease in cell count. Subsequently, through in vivo experiments on male ICR mice, we ascertained that SeNPs alleviated the detrimental impacts of PS-MPs on mouse liver. Results: SeNPs hinder the signaling pathway of NF-κB/NLRP3 inflammatory vesicles, which is crucial for reducing inflammation induced by PS-MPs. In terms of their mechanism, SeNPs hinder the abnormalities in mitochondrial fission, biogenesis, and fusion caused by PS-MPs and additionally enhance mitochondrial respiration. This enhancement is crucial in averting disorders in energy metabolism and inflammation. Conclusions: To summarize, the use of SeNPs hindered inflammation by regulating mitochondrial dynamics, thus relieving liver damage caused by PS-MPs in mice. The anticipated outcomes offer new research directions that can be referenced in terms of inflammatory injuries caused by PS-MPs.

## 1. Introduction

According to the relevant references, microplastics refer to plastic particles that have a diameter smaller than 5 mm [1,2,3]. Microplastics can be classified according to their sources, and as such can be divided into primary microplastics formed directly from industrial production and secondary microplastics produced by the decomposition of plastic-type waste [4,5]. In contrast, classification of microplastics can occur according to the polymer type, which includes polyethylene, polystyrene, polypropylene, polyethylene terephthalate, and polyvinyl chloride. As a novel environmental pollutant, microplastics have been found in rivers, oceans, the atmosphere, drinking water, and food sources around the world [6,7,8]. A variety of organisms can be exposed to microplastics through breathing, eating, and drinking, which accumulate in the organisms themselves and are passed up the food chain, so it is inevitable that humans will ingest microplastics [9]. The available references estimate that the amount of microplastics ingested by a person, through their digestive and respiratory tracts, to be between 74,000 and 121,000 microplastics over the course of a year [10]. Microplastic particles have been found in several organs of living organisms, posing serious health risks [11,12,13,14], while the accumulation of microplastics in the liver of mice has been widely reported [15,16,17].

In the body, the liver serves as the biggest gland involved in digestion and detoxification. The primary factors contributing to liver disease are drug toxicity, viral infections, and malnutrition. Chronic liver disease can be caused by a variety of factors, including hepatocellular injury, cell death, hypoxia, or irregular cellular metabolism, among which microplastic-induced hepatocellular injury due to oxidative stress in hepatocytes cannot be ignored. Patients with severe chronic liver disease may develop liver cancer or liver failure, or it may even result in death [18,19,20]. It has been demonstrated that PS-MPs activate the transcriptional machinery in the liver of Labrador dogs, leading to the enhanced expression of lipid-related genes [21]. PS-MP exposure significantly increased the inflammatory response in the liver, as demonstrated by the increased infiltration of Kupffer cells (KCs) and the increased expression of pro-inflammatory markers [22].

Selenium plays an important role in antioxidative stress and anti-inflammatory regulation and is regarded as a crucial micronutrient that is indispensable for human well-being [23]. Selenium was discovered by the chemist Betserius [24]. The necessity of Se for living organisms is due to the fact that it is a component of selenocysteine (Sec), found in various selenoproteins. Many selenoproteins have pronounced antioxidant activity, thereby playing a crucial role in the antioxidant defense of cells as well as the maintenance of cell redox homeostasis. The representative selenoprotein is GPx, which belongs to the antioxidant enzyme family and its active center is Sec. It mainly reduces hydrogen peroxide and lipid peroxides produced during metabolism by GSH and regulates the cellular microenvironment to realize the detoxification of the body. While excessive selenium can be harmful, the body needs a daily intake of small quantities of selenium to sustain physiological functions [25]. After being absorbed in the gut, selenium finally reaches the liver, which serves as the main site of selenium metabolism. It has also been shown that the liver takes up and absorbs selenium better than other tissues in order to utilize it [26]. Selenium provides the liver with a robust defense mechanism against hepatitis and aids during the prevention of fatty liver disease [27,28,29]. It should be noted that selenium exists in many forms, and SeNPs are made up of tiny selenium particles at the nanoscale that are more biologically active and less toxic than other forms of selenium [30,31]. Therefore, in this study, an animal model of nano-selenium antagonism induced by microplastic-induced liver injury was used as an example to simultaneously investigate the effects of microplastics and nano-selenium on mouse liver cells, so as to determine the specific mechanisms of nano-selenium and to propose new therapeutic targets against microplastics.

## 2. Materials and Methods

### 2.1. Microplastics

White powder PS-MPs, with a particle size of 5 μm, were purchased from Shuangfu Plasticising Trading Co (GuangZhou, China). The chemical nature of the PS-MPs was polystyrene and it was visually observed to be in the form of a powder; in order to make it mix uniformly during the liquid phase, the suspension of the PS-MPs was carried out in such a way to ensure they were dispersed uniformly in sterilized water using ultrasonic waves. Finally, the suspension containing the PS-MPs was stored at room temperature, protected from light and reconstituted weekly.

### 2.2. Nano-Selenium

After being mixed together, 1% Bacillus licheniformis, with OD600 = 0.5 and 20 mmol/L sodium selenite, was subjected to agitation in a shaker (180 r/min, 37 °C) for a duration of 4 days. Subsequently, the resulting mixture was centrifuged at a speed of 12,000 revolutions per minute, for a period of 20 min, in order to disintegrate the bacteria. After washing, the precipitate was ground in liquid nitrogen. The powdered substance was collected and rinsed three times using sterile, ultrapure water at a temperature of 4 °C, rotating at a speed of 4000 revolutions per minute, for a duration of 5 min. Following sonication for 15 min in an ice bath, the powder underwent three successive washes that were performed with 1.5 M Tris-HCl in 1% SDS (Ph = 8.3) at 4 °C and 12,000 rpm for 8 min. Subsequently, the next step also involved washing with sterile ultrapure water at 4 degrees Celsius and 10,000 rpm for 8 min. The cleaned sediment was reconstituted using sterile ultrapure water, followed by the addition of n-octanol in a volumetric specific suspension ratio of n-octanol of 2:1. It was mixed thoroughly and centrifuged at 4 °C, 12,000 r/min for 8 min. This washing process was repeated 4 times. Once a stable nano-suspension of nano-selenium was achieved, it was resuspended using an adequate quantity of sterile ultrapure water. After obtaining lyophilized nano-selenium, the liquid was discarded following a washing process, and the resulting precipitate was then stored at −80 °C for one day. The precipitate was then removed and freeze-dried in a freeze-dryer (−40 °C) for 72 h and finally stored at room temperature away from light.

### 2.3. Characterization of PS-MPs and SeNPs

A layer of conductive adhesive was glued onto the sample holder, and the freeze-dried lyophilized SeNP nanopowder or PS-MP powder was glued onto the conductive adhesive by gently picking a bit of it up with a toothpick. The unglued sample powder was removed with an ear wash ball, and the sample was placed into the sample chamber for evacuation, liquid nitrogen addition, and gold sprinkling. Then, a suitable area in the field of view was selected for spot analysis. Particle size and energy spectra of PS-MPs and SeNPs were determined using field emission electron microscopy (Techcomp, Shanghai, China).

The mixture of PS-MPs and sterile water and the mixture of SeNPs and sterile water were diluted to 50 mg/L and sonicated, followed by transfer to a zeta flow cell and a quartz cuvette cell (Malvern Panalytical, Malvern, UK) for characterization. The liquid-phase surface charge of PS-MPs and SeNPs was measured using zeta potential and a particle size analyzer (Malvern Panalytical, Malvern, UK).

### 2.4. In Vitro Experimental Design

Nano-selenium was mixed with deionized water in different ratios and sonicated to produce 0.5, 1, 2, 2.5, and 5 μmol/L solutions of nano-selenium. Microplastics were mixed with deionized water in different ratios and sonicated to produce 5, 50, 500, 1000, 1500 and 2000 μg/mL microplastic solutions. Normal mouse hepatocyte (NCTC1469) suspension was added to the proportionally configured medium containing 1 per cent horse serum and DMEM/HIGH dextros and incubated for 12 h in a constant-temperature (5% CO_2_, 37 °C) incubator (Panasonic) suitable for the survival of mouse hepatocytes. When the cell density reached 6.4 × 10^4^ per well in 6-well plates and 2.5 × 10^4^ per well in 96-well plates the next step was to treat them with different concentrations of PS-MPs and SeNPs.

#### 2.4.1. Culture of Mouse Hepatocytes

The cryopreservation tubes were removed from the liquid nitrogen and the cells were quickly thawed in a 37 °C water bath. As soon as the cells thawed, they were removed from the water bath and the outside of the cryopreservation tubes was quickly sterilized with an alcohol sprayer before placing them on an ultraclean bench. The thawed cell cryopreservation solution (1 mL) was pipetted into a 10 mL sterile EP tube, 4 mL of DMEM/HIGH GLUCOSE Complete Medium with 10% horse serum was added, and the mixture was centrifuged for 5 min at room temperature, 1000 r/min. After removing the supernatant, the precipitate was resuspended with 5 mL of DMEM/HIGH GLUCOSE Complete Medium with 10% horse serum. After it was mixed well, the cell suspension was added to T25 with a pipette gun and incubated at 37 °C under 5% CO_2_. During the incubation process, the solution was changed over 2–3 d. After the cells were fully grown, the test was carried out.

#### 2.4.2. Cell Growth Curve Determination

To ensure that cells exposed to microplastics and nano-sized selenium were in the equilibrium growth phase and normal, we measured the growth curves of hepatocytes. Hepatocytes (NCTC1469) in good growth condition were taken and digested with trypsinase, and the cells were resuspended by adding an appropriate amount of medium to prepare the suspension. After cell counting, the cell concentration was adjusted to (1~5) × 10^4^ cells/mL, and the cells were accurately inoculated into two 24-well plates with the same amount of cell suspension added to each well. Samples were taken at the time points of 0, 4, 12, 24, and 48 h, and the cells were counted with a biological inverted microscope (XZ-10, Mshot, Guangzhou, China) and a Watson hemocyte counter (177-112C, WATSON, Tokyo, Japan). Measurements at each time point were made three times and independent samples were taken in triplicate for cell number determination. Finally, taking the incubation time as the horizontal axis and the cell concentration as the vertical axis, the cell concentration obtained at each time point was marked on coordinate paper, and the cell growth curve was obtained by connecting each point to the line.

#### 2.4.3. Effect of PS-MPs on Hepatocyte Proliferation

Hepatocytes were counted using a biological inverted microscope (177-112C, WATSON, Tokyo, Japan) and Watson’s blood cell counter (Watson, Japan), and the cytosol was diluted in suspension to 10,000 cells/mL with DMEM/HIGH GLUCOSE Complete Medium with 10% horse serum. The bottom of the T25 bottles was observed with an inverted microscope and when the cells occupied 80–90% of the bottom area of the bottle, 10 µL of the cell suspension was withdrawn and placed in a 96-well plate. Subsequently, the culture was supplemented with 5 μm of PS-MPs at concentrations of 5, 50, 500, 1000, 1500, and 2000 μg/mL in the culture. Following prolonged incubation at a temperature of 37 °C for a duration of 6 h, 10 μL of CCK8 reagent was introduced into every well, and the incubation process was extended for an additional 2 h. Absorbance was measured at 450 nm using zymography (BGM LABTECH, Offenburg, Germany) and cell inhibition was determined from the data collected.

#### 2.4.4. Effect of SeNPs on Hepatocyte Proliferation

Based on the number of cells, the cell suspension was diluted to a concentration of 10,000 cells per milliliter. A volume of 10 μL of the cell suspension was withdrawn and introduced into 96-well plates. The bottom of the T25 flasks was observed using an inverted microscope, and when the area occupied by the cells reached 80–90% of the bottom area, selenium nano-solutions at concentrations of 0.5, 1, 2, 2.5, and 5 μmol/L were added to the cultures. Following a 12 h incubation period at a temperature of 37 °C, 10 μL of CCK8 solution was introduced into every well and left to incubate for a duration of 2 h. The measurement of absorbance was adjusted to 450 nm with zymography (BGM LABTECH, Germany) and the rate of cell proliferation was determined using the acquired data. We performed three biological replicates.

#### 2.4.5. ROS Detection

The ROS Assay Kit (Bioland Biologicals, Guangzhou, China) was used to detect ROS levels in liver NCTC1469 cells in real time. NCTC1469 cells were inoculated into 12-well plates. The NCTC1469 cells were wall-plated overnight and then divided into 4 groups with 3 wells per plate. They were blank group (Con), microplastic group with 100 μL of 1000 μg/mL PS-MPs added by a pipette gun, nano-selenium + microplastic group with 100 μL of 1000 μg/μL PS-MPs and 100 μL of 1 μmol/L SeNPs, and nano-selenium group with 100 μL of 1 μmol/L SeNPs. After 6 h of treatment, the groups were processed following the steps in the instruction manual and finally observed under an inverted fluorescence microscope (MF53, Guangzhou, China). Under the microscope, ROS emitted green fluorescence, and we quantified the extent of oxidative stress to determine whether ROS production was induced by converting each set of images into 8-bit images via ImageJ and then calculating gray values for the fluorescent areas.

### 2.5. In Vivo Experimental Design

All ICR mice studied were provided by Hunan Slaughter Jingda Laboratory Animal Co. Ltd. and housed at the Animal Experimentation Centre of the College of Animal Medicine, Huazhong Agricultural University, where this study complied with animal ethics requirements (SYxk2020-0084, WuHan, China). Thirty-six male mice (4 weeks old) were divided into six groups (n = 6), which were the control group, the low-dose PS-MP group (0.1 mg/kg/d MPs), the medium-dose PS-MP group (1 mg/kg/d MPs), the high-dose PS-MP group (10 mg/kg/d MPs), the group that was exposed to both PS-MPs and SeNPs (10 mg/kg/d MPs + 1 mg/kg/d SeNPs), and the nano-selenium group (1 mg/kg/d SeNPs). The low-dose PS-MP solution (20 mg/L), medium-dose PS-MP solution (200 mg/L), and high-dose PS-MP solution (2000 mg/L) were prepared with pure water and PS-MP powder before the experiment. Nano-selenium solution (200 mg/L) was prepared with pure water and nano-selenium powder. All animals were kept in the experimental animal house and were given sufficient food and water during the feeding period, which was 28 days. The mice were weighed weekly with a weighing scale and treated daily with PS-MP solution and SeNPs by gavage. After 28 days, all mice were euthanized by injection of 5% pentobarbital sodium (200 mg/kg). Blood samples were collected by cardiac puncture for biochemical analysis and liver tissue was extracted for subsequent experiments.

#### 2.5.1. H&E Staining of Liver

The mouse liver tissue was preserved in a 4% solution of paraformaldehyde for 24 h, followed by removal, dehydration in alcohol, and immersion in melted paraffin, and after the paraffin had solidified, it was trimmed for easy observation. Then, 5 μm sections of liver tissue embedded in paraffin were stained with hematoxylin and eosin. To make the sections transparent, the stained sections were dehydrated using anhydrous ethanol and subsequently washed with xylene. Ultimately, transparent portions were affixed with neutral resin and examined using a biological microscope (Ningbo Sunyu Instruments Co., Ltd., Ningbo, China).

#### 2.5.2. TEM Observation of Liver Sections

The mouse liver tissue was first treated with 2.5% glutaraldehyde in phosphate buffer for a duration of 12 h, followed by a subsequent 2 h fixation in 1% osmium acid. Additionally, the tissue samples underwent dehydration using a gradient of ethanol and were subsequently embedded in acetone. Samples were sectioned in an ultrathin sectioning machine. Finally, transmission electron microscope (TEM) images at different magnifications were taken under a transmission electron microscope (HITACHI, Tokyo, Japan).

#### 2.5.3. Blood Biochemical Analysis

Centrifuge tubes containing blood samples were placed in a 37 °C water bath for 5 min to produce the serum adequately. The serum was then placed in a high-speed refrigerated centrifuge (Centrifuge 5424R, Hamburg, Germany) at 4 °C, set at 4000 r/min, and centrifuged for 10 min. The BS-240VET (Mindray Biologicals Ltd., Shenzhen, China) was used to test serum biochemical parameters such as serum aspartate transaminase (AST), alanine aminotransferase (ALT), alkaline phosphatase (ALP), total bilirubin (T-BIL), direct bilirubin (D-BIL) and total bile acids (TBAs).

#### 2.5.4. Bone Densitometry Analysis

Mice were euthanized and placed in a bone densitometer (InAlyzer Republic, MEDIKORS, Suwon City, Republic of Korea) for measurement of bone mineral content (BMC), bone mineral density (BMD), fat content in tissue, and liver coefficient.

#### 2.5.5. Measurement of Antioxidant Indicators

Mouse serum malondialdehyde (MDA) levels, catalase (CAT), superoxide dismutase (SOD), and glutathione (GSH) activities were measured by assay kits (Nanjing Jiancheng Institute of Biological Engineering, Nanjing, China).

#### 2.5.6. Immunofluorescence Staining

Mouse paraffin sections were deparaffinized in xylene and washed with distilled water, then placed in a repair cassette with citrate antigen repair buffer (PH 6.0) in a microwave oven for antigen repair, after which they were washed with PBS (PH 7.4) buffer. After the sections were shaken dry, the sections were covered with 3% BSA and left for 30 min at room temperature. Next, ABclonal NLRP3 Rabbit pAb (A5652) and NFKB1 Rabbit pAb (A6667) as primary antibodies at a dilution of 1/200 were added dropwise to the sections and incubated at 4 °C overnight. After the sections were washed with PBS (PH 7.4), secondary antibody (iNOS Rabbit mAb (A3774)) at a dilution of 1/2000 was added dropwise to the sections, then they were incubated for 50 min at room temperature and covered with neutral gum, and, ultimately, the sections were scrutinized and pictures were taken with a fluorescent microscope.

#### 2.5.7. Quantitative Real-Time Quantitative Fluorescence PCR

The RNA was extracted from liver using Trizol reagent (Accurate Biology, Changsha, China) followed the manufacturer’s instructions. Then, the total RNA was reverse-transcribed into complementary DNA (cDNA) by referring to the reverse transcription kit instructions of Novizan (Nanjing, China). We used the reaction system of TSE202 2 × T5 Fast qPCR Mix (SYBR Green I) from Tsingke (Wuhan, China). An amount of 4 μL of cDNA was used in each qPCR reaction. The preincubation cycle ran at 95 °C once for 60 s, and the number of qPCR cycles was 40. The PCR analysis was conducted on the LightCycler96 (Roche) with the employment of QuantStudio™ 5 design and analysis software. The mRNA sequences of NF-κB, P-NF-κB, TNF-α, Cleaved Caspase1, Caspase1, NLRP3, IL-1β, DRP1, OPA1, MFN1, MFN2, PGC1-α, and SIRT1 were acquired from NCBI GeneBank. The primers were synthesized by Wuhan Tsingke Engineering. The calculation of relative gene expressions was performed by the method of 2^−△△Ct^ with the use of GAPDH as the internal reference gene.

#### 2.5.8. Western Blotting Analysis

Total proteins were extracted from liver tissues using RIPA lysate (Bi-osharp, Beijing, China), protein concentrations were determined by the Kaumas protein assay reagent, and the samples used were diluted to the same concentration by 0.9% NaCl for subsequent experiments. Next, the extracted protein samples were separated by 10% SDS-PAGE electrophoresis and subsequently transferred onto PVDF membranes. The membranes were soaked for 2 h in a mixture of skimmed milk and buffered saline (TBST) made from Tris and NaCl. Afterward, the PVDF filters were left to incubate overnight at a temperature of 4 °C with primary antibodies, which were diluted according to the provided guidelines. On the following day, the membranes underwent three washes with TBST solution for a duration of 5 min each. The membranes were then incubated overnight at 4 °C in secondary antibody diluted 1:8000 in TBST. Then, they were washed again in TBST 5 times for 5 min each at the end of the incubation. Both ECL A and ECL B were mixed in medium proportions in shaded centrifuge tubes, and PVDF membranes were placed in full contact with them and used for signal detection using a chemiluminescence imaging system (Nuclear Cheng Technology Development Co., Wuhan, China). All primary antibodies (NF-κB (p65), NLRP3, TNF-α, Caspase1, Cleaved Caspase1, P-NF-κB (p65), IL-1β, GAPDH) used in this section were from Wanlei bio and the secondary antibody was HRP-conjugated goat anti-rabbit IgG (H + L) (AS014) purchased from ABclonal (Wuhan, China).

## 3. Statistical Analysis

Data are reported as mean  ±  standard error of the mean (SEM) unless stated otherwise. Analysis of variance (ANOVA) was used for more than 2 groups. A *p*-value less than 0.05 was considered statistically significant. Statistical analysis and graphs were generated using GraphPad Prism 9.5 software: *: *p* < 0.05, **: *p* < 0.01, ***: *p* < 0.001, ns, not statistically significant; *p*-values of groups compared with each other: #: *p* < 0.05, ##: *p* < 0.01, ###: *p* < 0.001, ns: not statistically significant. All experiments were performed in 3 technical replicates. n = 6 for mouse experiments, n = 3 for cell experiments.

## 4. Results

### 4.1. Characterization of PS-MPs and SeNPs

According to the data presented in Figure 1A, the PS-MPs exhibit an average particle size of 5 μm and have a subspherical shape when observed using SEM. With TEM (Figure 1B), the SeNPs exhibit an approximately spherical shape with an average particle size of about 200 nm. Energy spectral scans were acquired for PS-MPs containing elements (Figure 1C,E) and SeNPs (Figure 1F,D) using SEM. The system is deemed to be highly stable if the magnitude of the zeta potential is equal to or greater than 30 mV, suggesting the presence of both electrostatic repulsion and spatial potential resistance among the particles [32]. The values of both PS-MPs (Figure 1G) and SeNPs (Figure 1H) are quite high. Zeta potential is a measure of the strength of the mutual repulsion or attraction between particles, and the higher the absolute value of zeta potential (positive or negative), the more the repulsive force exceeds the attractive force, which resists aggregation, and thus the more the system of selenium nanoparticles and microplastic solutions is stabilized.

### 4.2. In Vitro Tests

#### 4.2.1. Determination of the Growth Curve of Normal Mouse Hepatocytes

Growth curves were established for normal mouse NCTC1469 hepatocytes. Cells were counted from the start of recovery and the total number of cells was then measured at 24 h, 48 h, and 72 h. Cells from 0–24 h were in the lag phase, 24–48 h cells were in the stationary phase, and 48–72 h cells were in the equilibrium phase. Therefore, NCTC1469 cells were cultured for 48 h before exposure to PS-MPs and SeNPs was performed (Figure 2A).

#### 4.2.2. Detection of Impact of PS-MPs on Hepatocytes by CCK8

Following a 6 h exposure to PS-MPs, the viability of normal NCTC1469 hepatocytes from mice gradually declined as the concentrations of PS-MPs increased. Specifically, the survival of cells was significantly inhibited at 50 μg/mL (*p* < 0.05), 500 μg/mL (*p* < 0.01), 1000 μg/mL (*p* < 0.001), 1500 μg/mL (*p* < 0.01), and 2000 μg/mL (*p* < 0.01) of PS-MPs. The concentration of PS-MPs chosen for the subsequent study (Figure 2B) was 2000 μg/mL, as it was nearer to the IC50.

#### 4.2.3. Detection of Impact of SeNPs on Hepatocytes by CCK8

The impact of SeNPs on the viability of normal mouse NCTC1469 hepatocytes was observed for 6 h, as depicted in Figure 2C, at 0.5, 1, 2, 2.5, and 5 μmol/L. The findings indicated that small amounts of nano-selenium (0.5 μmol/L, 1 μmol/L) did not exhibit any harmful effect on mouse hepatocytes, and their cellular function was not notably distinct from that of the control group. However, when the concentration rose to 2 μmol/L, nano-selenium significantly stimulated the proliferation of hepatocytes (*p* < 0.05). Moreover, at a concentration of 2.5 μmol/L, their cellular activity displayed a highly significant difference (*p* < 0.01) compared to the control group.

#### 4.2.4. Results of ROS Assay in Mouse Hepatocyte Suspensions

Upon examination of normal mouse NCTC1469 hepatocytes using a fluorescence microscope, it was observed that the fluorescence intensity of mouse hepatocytes in the PS-MP group exceeded that of the control group. However, the fluorescence intensity of the PS-MP + SeNP group was reduced after the addition of SeNPs. The fluorescence intensity of the control and SeNP groups was weaker (Figure 2D).

### 4.3. In Vivo Tests

#### 4.3.1. Liver Dysfunction Induced by PS-MPs

Mouse serum biochemical parameters were measured in PS-MP-induced liver tissue (Figure 3A–F). The results of blood biochemical examinations indicated that the liver may be a target organ for PS-MPs. The expression levels of aspartate transaminase (AST), alanine aminotransferase (ALT), and serum alkaline phosphatase (ALP) were significantly higher in the exposed mice compared to the control. ALP was significantly increased in mice treated with 10 mg/mL of PS-MPs compared to the normal group. In addition, all other biochemical indices such as T-BIL, D-BIL, and TBA showed a significant increase in expression in a dose-dependent manner when compared to the control group.

#### 4.3.2. Bone Densitometry Results in Mice

After being exposed for 28 days, the mice were euthanized and subjected to testing using a bone densitometer (Figure 4A). A notable variation in bone mineral content (BMC) was observed in the low-dose PS-MP group compared to the control group (*p* < 0.05), while the remaining groups did not show any significant difference (Figure 4C). On the other hand, there was no notable variation in BMD among the groups, which did not show any statistical significance (Figure 4D). Nonetheless, there existed a notable disparity in the lipid composition within the mouse tissues among all PS-MP groups and the control group, particularly in the high-dose PS-MP group (10 mg/kg/d) in comparison to the control group (*p* < 0.001) (Figure 4B). In addition, the liver ratios of mice in the high-dose PS-MP group (10 mg/kg/day) showed significant differences (*p* < 0.01) relative to the control group (Figure 4E). Hence, the high-dose category of PS-MPs (10 mg/kg/d) was employed as the ideal dosage of PS-MPs in Animal Experiment II for subsequent examinations.

#### 4.3.3. SeNPs Reduce Liver Histopathological Abnormalities Caused by PS-MPs

In the H&E staining graphs of liver tissue sections (Figure 5A) it can be found that the control group showed normal hepatocyte morphology, with hepatocytes arranged in a spoke-like pattern, extending from the central vein in all directions, forming distinct hepatic cords. In contrast, the PS-MP group exhibited a markedly disorganized hepatocyte arrangement (yellow arrows), cytoplasmic vacuolization (red arrows), and inflammatory cell infiltration (green arrows). In the PS-MP + SeNP group, the histopathological results showed significant improvement compared to the PS-MP group. The cytoplasmic vacuolation was relatively attenuated, although there was still evidence of inflammatory cell infiltration. The SeNP group did not differ significantly from the control group. Liver sections exhibited normal hepatocytes with intact cytoplasm, as well as intact nuclei, nucleoli, and central veins.

TEM was used to observe the cellular ultrastructure in liver tissue 28 days after molding, as shown in Figure 5B. After being exposed to PS-MPs, hepatocytes displayed distorted nuclei (indicated by green arrows), blurred cell membrane boundaries (indicated by red arrows), and cytoplasm filled with vacuoles (indicated by blue arrows) when observed at a magnification of 5000×. At 15,000× magnification, PS-MP exposure showed a slightly abnormal mitochondrial structure with loosely arranged cristae (white arrows). The histopathological findings in the PS-MP + SeNP group, on the other hand, were significantly improved, with no deformation of the nucleus and clear cell membrane boundaries. Cell membrane boundaries were clearer in the control, SeNP and PS-MP + SeNP groups (yellow arrows). The SeNP group did not show any notable distinction when compared to the control group. The findings from all these experiments clearly demonstrated that SeNPs exert a notable counteractive impact on liver damage caused by PS-MPs.

#### 4.3.4. SeNPs Attenuates PS-MP-Induced Liver Function Impairment

In order to validate the protective effect of nano-selenium against liver injury caused by PS-MPs, we examined serum liver enzymes, such as aspartate transaminase (AST), alkaline phosphatase (ALP), and alanine aminotransferase (ALT), in accordance with histological findings. Levels of these enzymes indicative of liver injury were significantly elevated in the PS-MP group (10 mg/mL) compared to the normal group (*p* < 0.01). However, the increase was inhibited by SeNPs (*p* < 0.01) (Figure 6A–C). Furthermore, there was a notable rise in the fat content within the mouse tissues in the PS-MP group compared to the control group (*p* < 0.01). However, the PS-MP + SeNP group effectively mitigated the fat content increase caused by PS-MPs within the mouse tissues (*p* < 0.01) (Figure 6H).

#### 4.3.5. Protective Impact of Nano-Selenium on Oxidative Stress Caused by PS-MPs

The MDA content was significantly increased in the PS-MP-exposed group compared to the control group, while the elevated MDA content caused by PS-MPs was significantly reduced in the PS-MP + SeNP group (Figure 6F). In contrast, CAT (*p* < 0.01), GSH (*p* < 0.001), and SOD (*p* < 0.001) activities were significantly decreased in the PS-MP group compared to the control group. In contrast, the PS-MP + SeNP group significantly elevated the decrease in CAT, GSH, and SOD activities caused by PS-MPs (Figure 6D,E,G). These results suggest that PS-MP exposure led to an imbalance of the redox system in mouse liver tissues, resulting in oxidative damage.

#### 4.3.6. Effect of SeNPs on PS-MP-Induced Liver-Injury-Related Proteins Through Regulation of the NF-κB/NLRP3 Inflammatory Pathway

NF-κB in the PS-MP group was mainly expressed in the central vein, while NLRP3 was mainly expressed in the cells surrounding the central vein (Figure 7A) The protein expression levels of NF-κB, P-NF-κB, TNF-α, Cleaved Caspase1, Caspase1, NLRP3, and IL-1β were significantly higher in the PS-MP group than in the control group (*p* < 0.01). The protein expression levels were all significantly lower in the PS-MP + SeNP group compared to the PS-MP group (*p* < 0.01) (Figure 7B,C). These results suggest that SeNPs attenuated the liver inflammatory injury induced by PS-MPs by regulating the NF-κB/NLRP3 inflammasome pathway.

#### 4.3.7. Effects of SeNPs on Genes Related to PS-MP-Induced Liver Injury Through Regulation of the NF-κB/NLRP3 Inflammatory Vesicle Pathway

The gene expression levels of NF-κB, TNF-α, Caspase1, NLRP3, and IL-1β were significantly higher in the PS-MP group than in the control group (*p* < 0.01). The gene expression levels were all significantly lower in the PS-MP + SeNP group compared to the PS-MP group (*p* < 0.01) (Figure 8A–E). These results suggest that nano-sized selenium was able to attenuate the inflammatory liver injury induced by PS-MPs.

#### 4.3.8. SeNPs Attenuate PS-MP-Induced Mitochondrial Dysfunction

Compared with the control group, the mRNA levels of mitochondria-related genes, such as OPA1, DRP1, MFN1, and MFN2, were significantly higher in the PS-MP group. (Figure 8F–K), indicating accelerated mitochondrial division and fusion. Among them, MFN1 is involved in the attachment of nearby mitochondria, maintenance of mitochondrial membrane potential, and apoptosis. MFN2 is a multifunctional protein located in the outer mitochondrial membrane, controlling mitochondrial fusion and physical interactions with other organelles. However, after co-treatment with SeNPs, the mRNA levels of all of them were significantly down-regulated. Furthermore, taken together, these results suggest that SeNPs have a protective effect on mitochondria.

## 5. Discussion

Accumulation of microplastics causes oxidative stress, inflammation, and apoptosis-related toxic effects in the liver, as well as disturbances in glucose and lipid metabolism in the liver [33]. The liver plays a crucial role in the storage, conversion, and breakdown of medications [34]. Several studies have been conducted on the mammalian biosafety of PS-MPs [35]. However, studies on the protective effects and molecular mechanisms of combined SeNPs against PS-MP-induced liver injury have not been reported. This study has confirmed for the first time that SeNPs reduce liver damage caused by microplastics by regulating inflammation mediated by the NF-κB/NLRP3 inflammatory pathway. This study confirmed that nano-selenium effectively reduced liver damage induced by PS-MPs, as observed through histopathological examination and serum biochemical analysis. Additionally, it unveiled the precise molecular mechanisms of nano-selenium in terms of mitochondrial dynamics, oxidative stress, and inflammation.

First, we determined in vitro experiments with normal mouse NCTC1469 hepatocytes that PS-MPs have a damaging effect on hepatocytes, while SeNPs have a proliferative effect on hepatocytes. The mitochondria primarily metabolize oxygen in aerobic organisms. Reactive oxygen species (ROS) are typically generated by organisms during cellular respiration. They are primarily found in liver cells and immune cells, where they promote cellular maturation and programmed cell death [36,37]. Excessive generation of ROS can result from different cellular stresses, and prolonged exposure to elevated levels of ROS is a significant factor contributing to numerous pathological conditions [38,39,40,41]. The findings of this study align with previous research on the harmful effects of microplastics on various tissues. This indicates that the inflammation caused by microplastics is strongly linked to the excessive buildup of ROS, which triggers inflammatory signaling and leads to inflammatory damage [42]. On the other hand, selenium (Se) acts as a strong catalyst for glutathione peroxidase (GPx) [43]. GPx, an important selenoprotein within the body, plays a crucial role in regulating the excessive generation of radicals in areas of inflammation. It also ensures the maintenance of the intracellular redox equilibrium and provides antioxidant properties to safeguard healthy cells against oxidative harm caused by ROS. Furthermore, studies have demonstrated that the augmentation of Se-GPx activity in the livers of mice leads to an improved capacity of the antioxidant system to safeguard against harm [44]. The balance between oxidants and antioxidants is controlled by the activity of Se-GPx [45]. Nano-selenium has strong hepatoprotective bioactivities and may be an antioxidant treatment [46]. Moderate amounts of nano-selenium in vivo are effective in ameliorating liver injury induced by PS-MPs. Therefore, nano-selenium supplementation can attenuate oxidative stress (ROS) induced by PS-MP exposure.

To summarize, our experiments conducted in a controlled environment demonstrated that PS-MPs increase the levels of reactive oxygen within cells, while the inclusion of SeNPs mitigated the harm inflicted on liver cells by PS-MPs. Based on this foundation, we performed in vivo tests on ICR mice using varying doses of PS-MPs. A previous study showed that polystyrene microplastics induced the formation of extracellular traps in macrophages, activating the ROS/TGF-β/Smad2/3 signaling axis, leading to an inflammatory response and epithelial–mesenchymal transition in hepatocytes [47]. The impairment of hepatocytes can result in the secretion of diverse enzymes and a prompt corresponding response, indicating significant liver damage [48]. In this experiment, several biochemical indices related to mouse liver were evaluated by oral administration of PS-MPs. These indicators play a crucial role in the metabolism of mammalian liver cells. The results of the test showed that there were significant differences in the indicators related to liver damage as the concentration of PS-MPs increased and that there was accumulative effect of concentration. The results showed that exposure to different concentrations of PS-MPs caused an increase in various biochemical parameters of liver function in mice, as well as an increase in tissue, fat content, and liver coefficients, indicating that liver cells were damaged.

Therefore, among the animal tests with different PS-MP concentrations, we chose the concentration of PS-MPs with the most significant liver damage (10 mg/kg/d) for a further test to explore the combination with SeNPs. The results showed that the levels of several antioxidant enzymes in the serum of mice were significantly decreased after exposure to PS-MPs, while the levels of MDA increased significantly. Among them, GSH plays an important role in scavenging reactive oxygen species, SOD plays a role in scavenging free radicals, and CAT plays a role in scavenging hydrogen peroxide in the body [49]. MDA is a product of lipid peroxidation, which can be used as an indicator of oxidative damage [50]. These results suggest that exposure to PS-MPs decreases the level of antioxidant defenses in mice, leading to increased oxidative stress. However, the impairment of hepatic oxidative stress by PS-MPs was significantly reversed by the addition of SeNPs.

Meanwhile, a study showed that exposure to PS-MPs induced apoptosis and pyroptosis in granulosa cells in rat ovaries, leading to an increase in inflammatory markers, and the mechanism of action was associated with activation of the NLRP3/Caspase1 pathway [51]. A study of microplastics in brain injury also confirmed that microplastics disrupt mitochondrial function and activate AMPK signaling, leading to inflammation in the brain. This evidence suggests that microplastics damage tissues through different mechanisms and that elevated inflammatory markers in the liver are a sign of the damage they cause [52,53,54]. Aberrant expression of NF-κB pathway pro-inflammatory cytokines is a major feature of the inflammatory response [55,56,57].

The findings indicated that PS-MPs activate the NF-κB pathway in the mouse liver, functioning as a pivotal hub for signaling, an emergency sensor, and being responsive to nearly all hazardous cues [58]. Moreover, it was evident that PS-MPs stimulate NLRP3 inflammatory vesicles within the liver. NLRP3, one of the NOD-like receptors, stands out as the most adaptable and medically important inflammatory vesicle, triggered by a diverse range of stimuli that are not commonly seen [59]. Increased mitochondrial cleavage has been reported to activate the generation of NLRP3 inflammatory vesicles [60,61]. In addition, NF-κB promotes the synthesis of mRNAs for several inflammation-related genes, including NLRP3 [62,63]. The findings indicate that PS-MPs have the ability to trigger inflammation in NLRP3 vesicles, leading to scorch-induced cell death. In addition, activation of the NF-κB pathway may be one of the mechanisms of NLRP3 inflammatory vesicle initiation. Activation of NLRP3 inflammatory vesicles initiates the NF-κB pathway and mediates cleavage. NLRP3 inflammatory vesicles can be activated by inflammatory factors, mitochondrial reactive oxygen species, and lysosomal enzyme damage [64,65,66]. The NF-κB pathway [67] can be activated by oxidative stress, resulting in the release of inflammatory factors and ultimately leading to the occurrence of NLRP3 inflammatory-vesicle-induced scorch death. Thus, our hypothesis is that the NF-κB signaling pathway could be activated and NLRP3 inflammatory-vesicle-induced scorch death could be mediated by PS-MP-induced oxidative stress. Although the precise mechanism is still unknown, this offers a solid foundation for conducting exposure studies on PS-MPs.

Mitochondria play a role in multiple crucial cellular functions, and excessive exposure to PS-MPs disrupts the dynamics of mitochondria, ultimately resulting in the death of liver cells in mice. SIRT1 positively regulates pGC1-α, a crucial mediator of mitochondrial biogenesis, as well as the fusion and fission processes of mitochondria [68]. The findings of this test indicate that PS-MPs lead to increased mitochondrial fragmentation, decreased merging and formation, and ultimately oxidative stress and apoptosis by enhancing mRNA expression of multiple genes associated with mitochondria [69,70]. SeNPs have been found to improve the function of the mitochondrial electron transport chain and control the production of new mitochondria [71,72]. The findings indicate that SeNPs have the ability to preserve the balance of mitochondrial dynamics by controlling the proteins responsible for both mitochondrial division and merging.

It is important to take into account various restrictions in our research. For instance, our attempts to delve deeper into the involvement of oxidative stress in the activation of NLRP3 inflammatory vesicles were unsuccessful. In our present investigation, we solely investigated the mechanisms linked to NF-κB/NLRP3 induction in liver cells due to exposure to PS-MPs, without analyzing the correlation between alternative programmed cellular pathways and diverse forms of cell death. Our future studies will center around these areas.

## 6. Conclusions

The current research illustrates that PS-MPs exhibit cumulative impacts at varying levels of exposure. Oral administration of PS-MPs resulted in the activation of the NF-κB/NLRP3 inflammatory signaling pathway in the liver of mice, resulting in a certain degree of liver damage. Mitochondrial pathway activation caused oxidative stress, inflammation, and hepatocyte scorching, resulting in hepatocyte damage, liver dysfunction, and hepatotoxicity. On the other hand, the inclusion of SeNPs greatly reduces the liver damage caused by microplastics. Nano-selenium, a trace element, is crucial in the pathophysiological mechanisms of organisms. The data and references presented in this study can be used to evaluate the biological hazard of orally consuming environmentally exposed PS-MPs in mice. The findings of the research could additionally aid in the advancement of policies aimed at preventing and managing plastic pollution.

## Figures and Tables

**Figure 1 nutrients-16-03878-f001:**
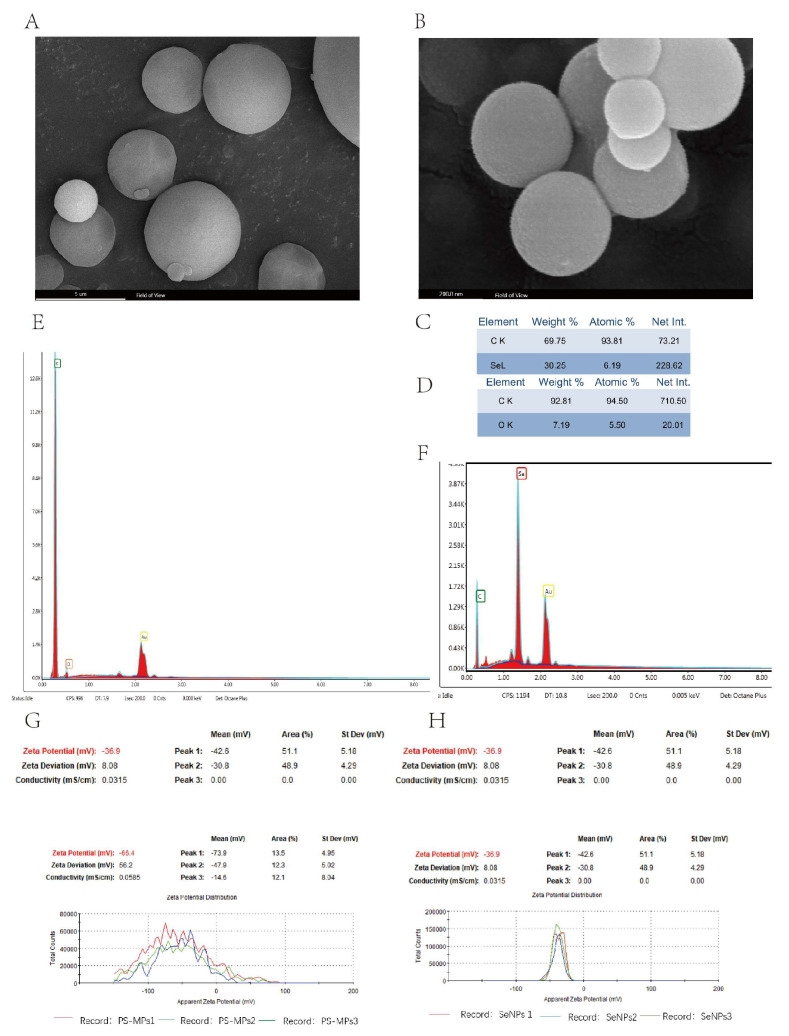
Characterization results of PS-MPs and SeNPs. (**A**) Scanning electron microscopy of PS-MPs. Scale bar, 5 μm. (**B**) Scanning electron microscopy of SeNPs. Scale bar, 200 nm. (**C**,**E**) EDS of PS-MPs. (**D**,**F**) EDS of SeNPs. (**G**) ζ-potential map of PS-MPs. (**H**) ζ-potential diagram of SeNPs.

**Figure 2 nutrients-16-03878-f002:**
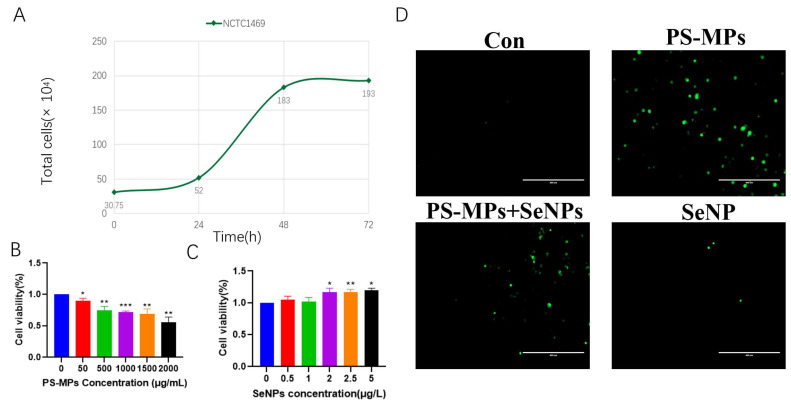
Effects of PS-MPs and SeNPs on mouse NCTC1469 hepatocyte lines. (**A**) Growth curves of normal mouse NCTC1469 hepatocyte cell suspensions, (**B**) cell survival rates after 6 h of action of different PS-MP concentrations (n = 3), (**C**) cell survival rates after 6 h of action of different SeNP concentrations (n = 3), (**D**) ROS fluorescence pictures of the groups. Scale bar, 400 μm. *: *p* < 0.05, **: *p* < 0.01, ***: *p* < 0.001.

**Figure 3 nutrients-16-03878-f003:**
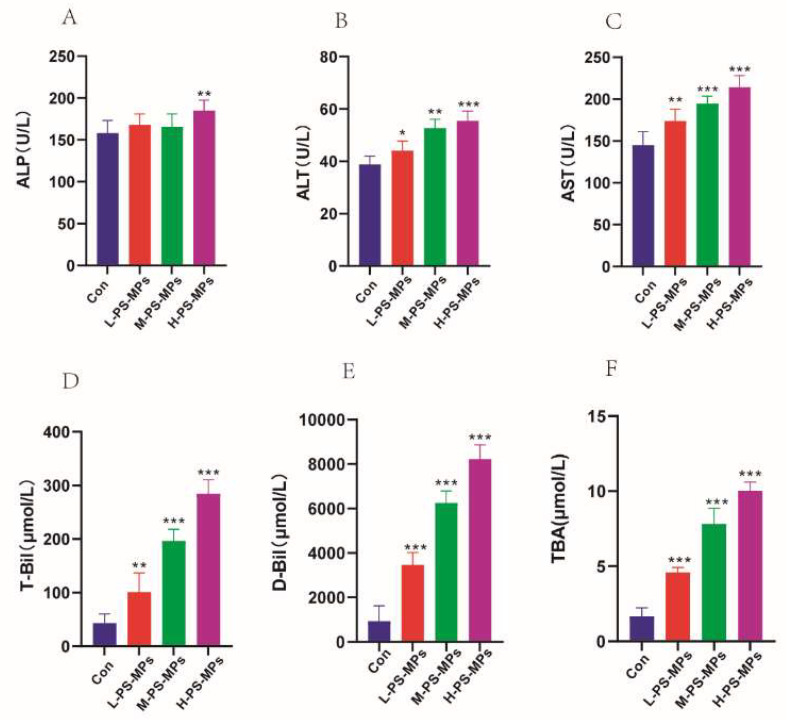
Effects of PS-MPs on biochemical indices in mice (n = 6). (**A**) Serum alkaline phosphatase (ALP), (**B**) serum alanine aminotransferase (ALT), (**C**) serum aspartate transaminase (AST), (**D**) total bilirubin (T-Bil), (**E**) direct bilirubin (D-Bil), (**F**) total bile acid (TBA). *: *p* < 0.05, **: *p* < 0.01, ***: *p* < 0.001.

**Figure 4 nutrients-16-03878-f004:**
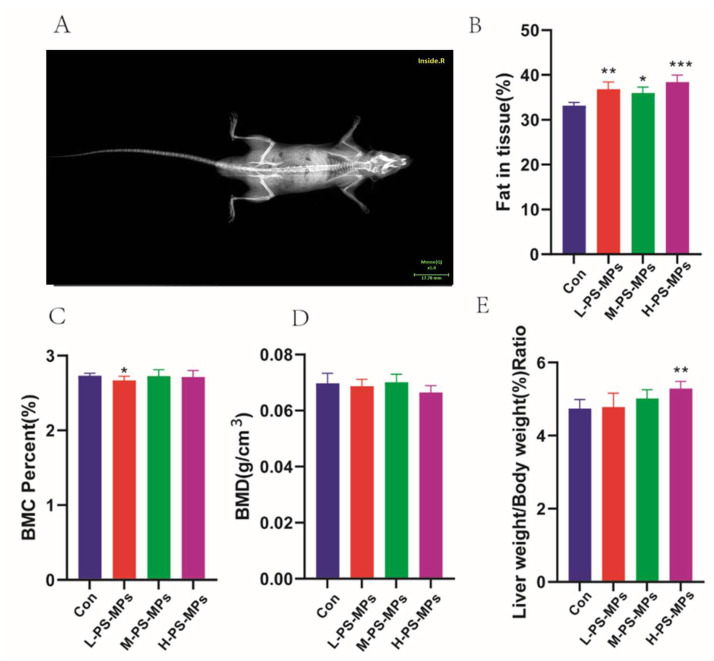
Radiographic images of mouse bone densitometry (n = 6). (**A**) Bone mineral density map (n = 6), (**B**) fat content in tissue (n = 6), (**C**) bone mineral content (BMC) (n = 6), (**D**) bone mineral density (BMD) (n = 6), (**E**) liver coefficient (n = 6). *: *p* < 0.05, **: *p* < 0.01, ***: *p* < 0.001.

**Figure 5 nutrients-16-03878-f005:**
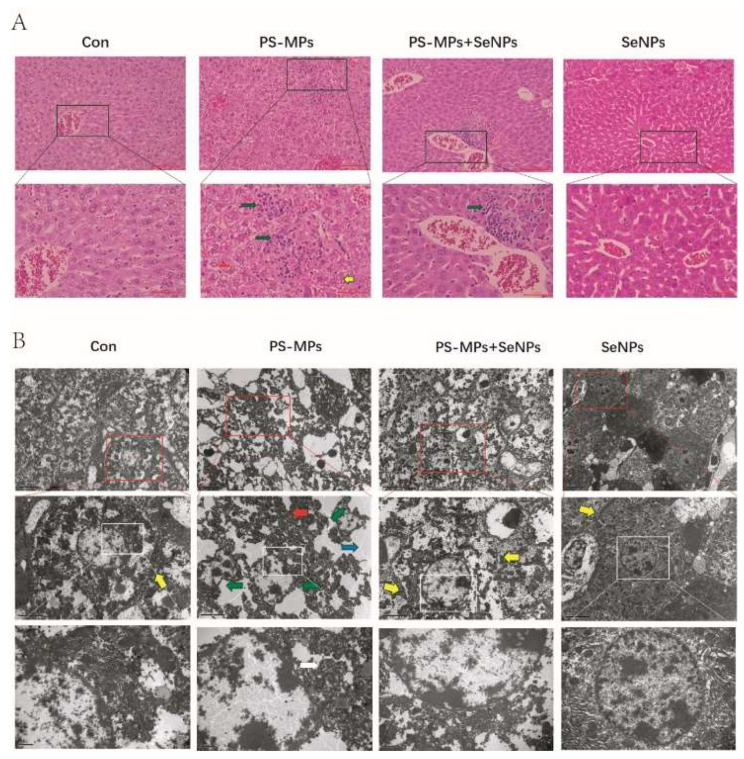
H&E stain images and electron microscopic images of liver tissue sections. (**A**) H&E stain map of mouse liver tissue. The overall magnification is 200× while the magnification of the black boxed section is 400×. Green arrows, inflammatory cell infiltration; yellow arrows, cytoplasmic vacuolation. (**B**) Transmission electron micrographs of mouse liver tissue. Red arrows: blurred cell membrane boundaries; yellow arrows: clear cell membrane boundaries; green arrows: distorted nuclei; white arrows: slightly abnormal mitochondrial structure with loosely arranged cristae; blue arrows: cytoplasmic vacuolation.

**Figure 6 nutrients-16-03878-f006:**
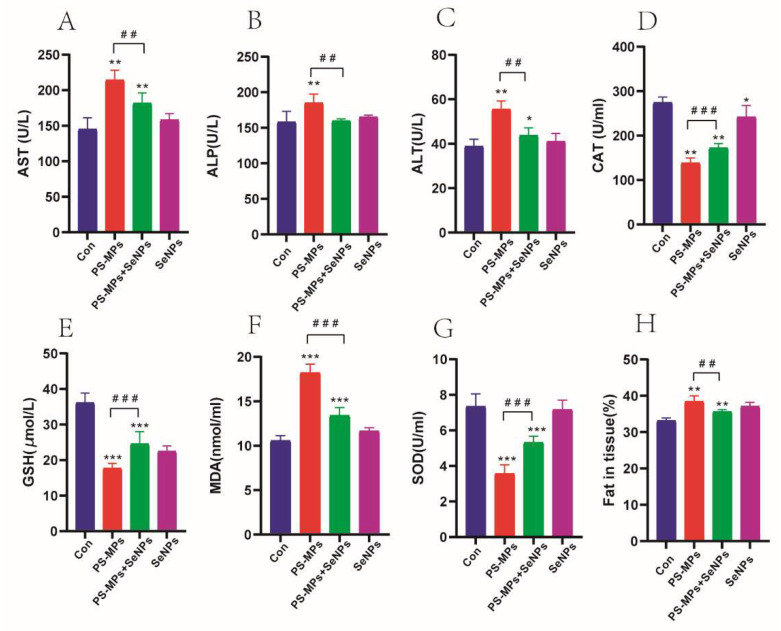
SeNPs attenuate PS-MP-induced liver function impairment. (**A**) Serum aspartate transaminase (AST) (**B**) serum alkaline phosphatase (ALP), (**C**) serum alanine aminotransferase (ALT), (**D**) catalase (CAT), (**E**) glutathione (GSH) (**F**) malondialdehyde (MDA), (**G**) superoxide dismutase (SOD), (**H**) fat content in tissues. The data are presented as the average plus or minus the standard deviation (n = 6). *: *p* < 0.05, **: *p* < 0.01, ***: *p* < 0.001. *p*-values of groups compared with each other: ##: *p* < 0.01, ###: *p* < 0.001.

**Figure 7 nutrients-16-03878-f007:**
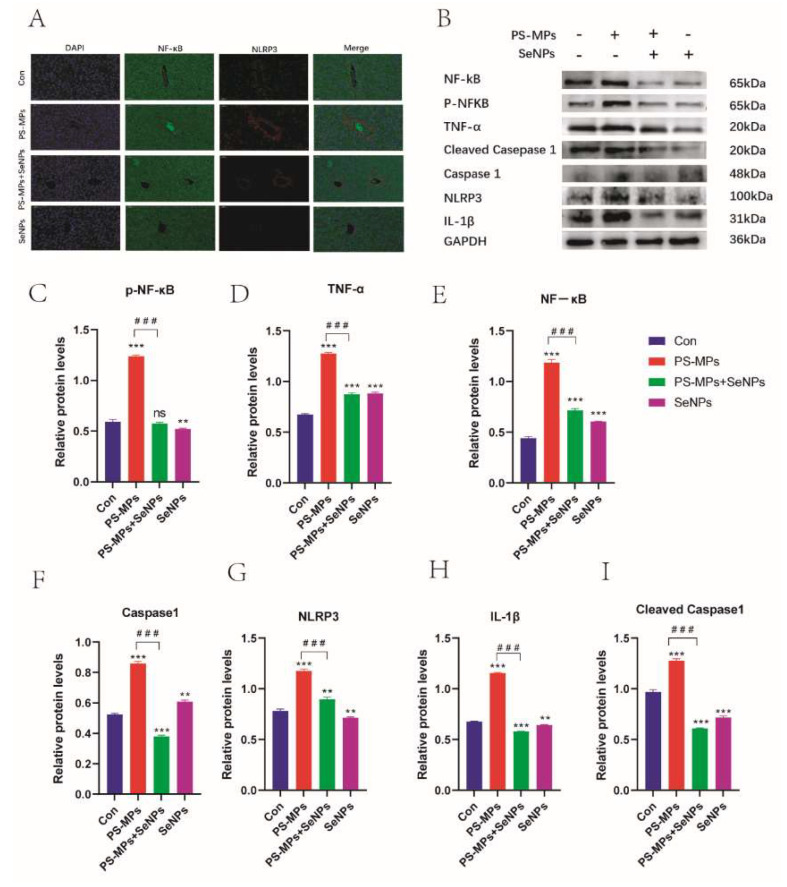
Effects of SeNPs on PS-MP-induced liver-injury-related proteins by modulating the NF-κB/NLRP3 inflammatory vesicle pathway. (**A**) Immunofluorescence expression of NF-κB protein and NLRP3 protein in the liver of each group of mice. (**B**) Protein blots of NF-κB, P-NF-κB, TNF-α, Cleaved Caspase1, Caspase1, NLRP3, IL-1β (n = 3). (**C**–**I**) Quantitative expression of these proteins (n = 3). **: *p* < 0.01, ***: *p* < 0.001, ns: not statistically significant. *p*-values of groups compared with each other: ###: *p* < 0.001.

**Figure 8 nutrients-16-03878-f008:**
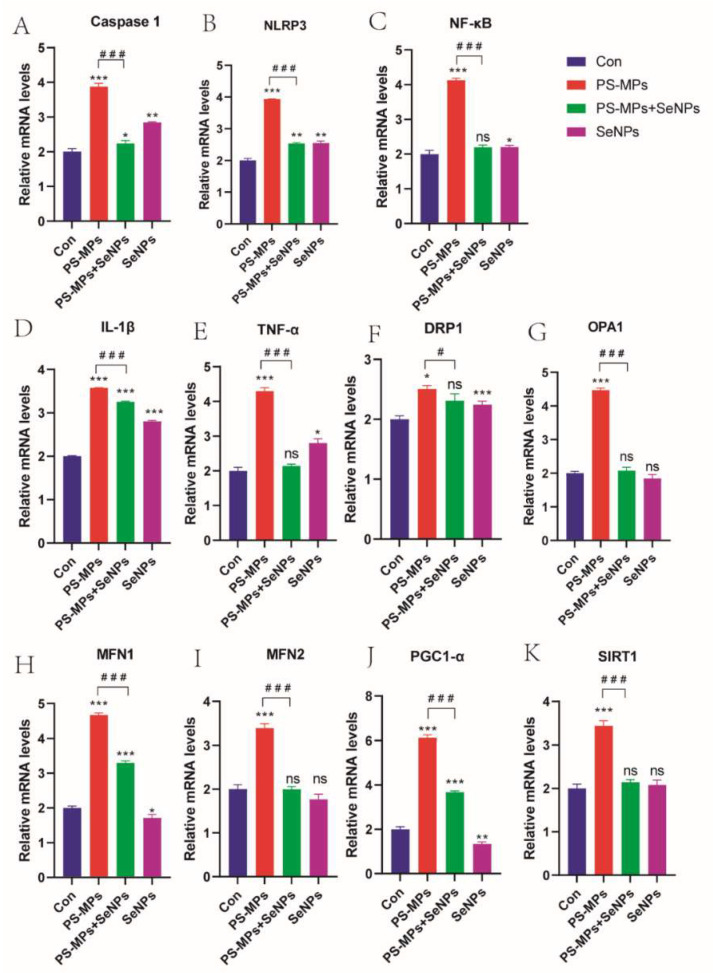
Measurement using quantitative real-time fluorescence qPCR. (**A**–**E**) Levels of gene expression related to inflammation. (**F**–**K**) Levels of gene expression linked to mitochondria. *: *p* < 0.05, **: *p* < 0.01, ***: *p* < 0.001, ns: not statistically significant. *p*-values of groups compared with each other: #: *p* < 0.05, ###: *p* < 0.001. ns: not statistically significant.

## Data Availability

The original contributions presented in the study are included in the article, further inquiries can be directed to the corresponding author.

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
