# Peer review of "Nano-Selenium Modulates NF-κB/NLRP3 Pathway and Mitochondrial Dynamics to Attenuate Microplastic-Induced Liver Injury"

_nutrients, 2024, doi:10.3390/nu16223878_

Round 1

Reviewer 1 Report (Previous Reviewer 4)

Comments and Suggestions for Authors

Comments to the Authors of manuscript number nutrients-3285788 entitled “Nano-selenium modulates NF-κB/NLRP3 pathway and mitochondrial dynamics to attenuate microplastic-induced liver injury”

1.In the entire manuscript, please check the spelling.-µ -a typographical error

2. 8-9-to investigate the efficacy of nano-selenium (SeNPs) in mitigating liver damage caused by microplastics

3. 26-29-The introduction lacks citations for the definitions and classifications provided.

4. 35-36-statistics or references

5. 44-45-which factors are particularly relevant in the context of microplastics.

6. 55-57-examples of specific selenoproteins and their functions are needed

7. 150-152- how the cells are initially counted and prepared for dilution?

8. 158-Zymography typically refers to a technique used to detect enzyme activity.

9. 161-how the initial cell count is performed?

10. 176-The concentration of PS-MPs seems unusually high without justification. There should be a reference or rationale for choosing this concentration.

11. 182-Quantitative analysis should be the primary method, and it should describe how ImageJ was specifically used.

12. 185-the age, weight, and sex distribution of the mice to understand the sample population

13. 188-197-why these specific doses were selected and their potential relevance to real-world exposure levels

14. 231-which specific kits were used or their manufacturer details

15. 261-272-how the protein concentrations were determined before loading the samples on the gel

16. 279-282-what specific comparisons were made, post-hoc tests?

17. 288-how this shape was determined?

18. 290-an approximately spherical shape

19. 296-The mechanism by which high zeta potential prevents clumping should be briefly explained or referenced.

20. 303-NCTC1469 refers to a cell line, not hepatocytes specifically from mice

21- 306-307-what criteria were used to determine these phases?

22. 318-how the IC50 was determined for PS-MPs?

23. 329-which concentrations were used?

24. 343-The term "expression" is typically used in the context of gene expression. For biochemical indices, it would be more appropriate to use "levels" or "concentrations."

25. 351-352-It is unclear what "BMC content" refers to

26. 373-374-Quantitative data?

27. 409-410-the concentration of SeNPs used?

28. 417-418-quantitative data for MDA?

29. 431-432-This conclusion is valid based on the presented data

30. 449-451-Including specific gene names would clarify which mitochondrial processes are being impacted.

31. Discuss specific mechanisms in more depth.

Author Response

We are very grateful to your and the reviewers’critical comments and thoughtful suggestions. Based on these comments and suggestions, we have made careful modification on the original manuscript. All changes made to the text are in yellow in the revised manuscript so that they may be easily identified. All of your questions were answered below.

Comments 1:[ In the entire manuscript, please check the spelling.-µ -a typographical error]

Response 1:[ Thank you very much for bringing this to our attention. We apologize for any errors caused by revising the article. We have corrected all instances of "μ" in the word and also corrected the spelling error in the unit symbol.]

Comments 2: [8-9-to investigate the efficacy of nano-selenium (SeNPs) in mitigating liver damage caused by microplastics]

Response 2:[We sincerely appreciate the valuable comments. But we are sorry that we did not understand your specific meaning. We have mentioned the role of nano-selenium and the harm of microplastics at the same time in line 8-9. We look forward to the follow-up reply, which can help us to better improve the quality of this paper]

Comments 3:[ 26-29-The introduction lacks citations for the definitions and classifications provided.]

Response 3:[As suggested by the reviewer, we have added references to support the definitions and classifications ]

Comments 4:[35-36-statistics or references]

Response 4:[ Thank you very much for your suggestion, we have added the relevant data to the manuscript]

Comments 5:[ 44-45-which factors are particularly relevant in the context of microplastics.]

Response 5:[ Thank you very much for your suggestions, they are very helpful for us to improve the quality of our papers. We have already emphasized the correlation between microplastics and hepatocyte injury.]

Comments 6:[ 55-57-examples of specific selenoproteins and their functions are needed]

Response 6:[ Thank you very much for your advice. Followed your tips to make changes to the content to make my article more informative. We have added classic selenoprotein related descriptions to the manuscript.]

Comments 7:[ 150-152- how the cells are initially counted and prepared for dilution?]

Response 7:[ Thank you very much for your question. Our method for initial counting of hepatocytes is described in 2.4.2. Also in response to your question we add at 2.4.3. The method is as follows: Hepatocytes were counted by bio-logic inverted microscope and Watson's blood cell counter, and the cytosol was diluted in suspension to 10,000 cells/ml with DMEM/HIGH GLUCOSE Complete Medium with 10% horse serum.]

Comments 8:[ 158-Zymography typically refers to a technique used to detect enzyme activity.]

Response 8:[ Thank you very much for asking the question. It is true that enzyme markers are commonly used for the detection of enzyme activity. However, after adding cck8 reagent to a cell suspension, the concentration of the cell suspension can also be detected by measuring its absorbance with an enzyme marker. The principle is that CCK-8 can be reduced by the dehydrogenase enzyme in the cell mitochondria to produce a highly water-soluble orange-yellow dirty product. The shade of color is proportional to cell proliferation and inversely proportional to cytotoxicity. The OD value is measured at 450 nm using an enzyme marker, indirectly reflecting the number of viable cells.]

Comments 9:[ 161-how the initial cell count is performed?]

Response 9:[ Thank you very much for your question. The method of initial cell counting is described in 2.4.2]

Comments 10:[ 176-The concentration of PS-MPs seems unusually high without justification. There should be a reference or rationale for choosing this concentration.]

Response 10:[ Microplastic concentrations vary in different regions or environmental areas around the globe, and rather than modelling realistic microplastic concentrations in specific environments, our study focuses on exploring the explicit effects of microplastics on organismal tissues. At the same time, studies have shown an exponential increase in the production of plastics in humans between 2004-2014, making our chosen dose somewhat predictive [1]. Although the environmental dose of 1 mg/kg is larger than common doses, there are still many studies using this dose to explore the effects of microplastics on various tissues and organs of organisms [2,3]. In our preliminary experiments, we found that MPs at a concentration of 1 mg/kg did not readily agglutinate and precipitate, which ensured that the amount of microplastics ingested by each mouse was close to the same amount. Also to ensure consistency between in vitro and in vivo studies, we intend to use different gradients of microplastic concentration to observe the cellular response to them. There have been studies where THP-1 cell lines were treated with microplastic suspensions ranging from 1-1000ug/ml to assess their toxicity to the cells[4]. It has also been shown that concentrations of 10 μg/mL (5–200 µm), had an adverse effect on cell viability, and 20 μg/mL (0.4 µm) on cytokine release.[5]]

Comments 11:[ 182-Quantitative analysis should be the primary method, and it should describe how ImageJ was specifically used.]

Response 11:[ Thank you very much for your suggestion, we have added the quantitative method for ROS staining by image J software to the manuscript.]

Comments 12:[ 185-the age, weight, and sex distribution of the mice to understand the sample population]

Response 12:[ Thank you very much for your suggestion. We are using 4-week-old, male ICR mice with a weight distribution of around 20g. The above is described in subsection 2.5.]

Comments 13:[ 188-197-why these specific doses were selected and their potential relevance to real-world exposure levels]

Response 13:[ Thank you very much for your valuable questions. We have responded to a portion of it in comments 10. And we have a similar point of view on mice. Concentrations of microplastics vary in various regions or environmental areas around the globe, and rather than modelling the concentration of microplastics in a particular environment in reality, our study focuses on exploring the definitive effects of microplastics on the tissues of living organisms. At the same time, studies have shown that human plastic output increased exponentially between 2004-2014, so the dose we chose is somewhat predictive[6]. Although 1mg/kg is a larger than common environmental dose, there are still many studies that use this dose to explore the effects of MPs on various tissues and organs of organisms[7 ,8]. In our pre-experiment, we found that a concentration of 1mg/kg of MPs is less prone to agglutination and precipitation, which ensures that the amount of microplastics ingested by each mouse is close to the same amount.]

Comments 14:[ 231-which specific kits were used or their manufacturer details]

Response 14:   [Thank you very much for your question. We use MDA kit, CAT kit, SOD kit and GSH kit from Nanjing Jiancheng Institute of Biological Engineering Company to test the above indexes.]

Comments 15:[ 261-272-how the protein concentrations were determined before loading the samples on the gel]

Response 15:[ Thank you very much for your suggestion. After extracting the proteins, we measured the concentration of the proteins at 280 nm absorbance by the BCA assay of the Kaumas protein assay reagent, and finally diluted them to the same concentration for subsequent experiments.Thanks toWeWe have included theWe have made thisWe have added thisWe have added this to theWe have added this to the manuscript]

Comments 16:[ 279-282-what specific comparisons were made, post-hoc tests?]

Response 16:[ Thank you very much for asking the question. However, we did not understand what your question was trying to convey, and we describe the data analysis methods used for this experiment in lines 279-282.]\

Comments 17:[288-how this shape was determined?]

Response 17:[ Thank you very much for your question. We are scanning the microplastic powder and selenium nanopowder by electron microscope, in the image, both microplastic and selenium nanopowder are in spherical shape. The content has been mentioned in the manuscript.]

Comments 18:[290-an approximately spherical shape]

Response 18:[ Thank you very much for your meticulous review. What we have here is due to the fact that it is not possible to tell under an electron microscope whether the powder is perfectly spherical or not, and whether it is chipped or not, so we describe it as approximately spherical]

Comments 19:[296-The mechanism by which high zeta potential prevents clumping should be briefly explained or referenced.]

Response 19:[ Thank you very much for your suggestion, we have explained the significance of zeta potential values for solution stability.]

Comments 20:[303-NCTC1469 refers to a cell line, not hepatocytes specifically from mice]

Response 20:[ Thank you for your question. Because 4.2.1 is part of the results of the cellular assay, NCTC1469 is a mouse normal liver cell line commonly used in toxicology studies.]

Comments 21:[ 306-307-what criteria were used to determine these phases?]

Response 21:[ Thank you very much for your question. After the cells reach saturation density, they stop growing, enter the flat-top phase, and then degenerate and die. In order to accurately describe the dynamic changes in cell number during the whole process, a typical growth curve can be divided into four parts: the latent phase with slow growth, the exponential growth phase with a large slope, the flat-topped phase with a platform shape, and the degenerative decay.]

Comments 22:[318-how the IC50 was determined for PS-MPs?]

Response 22:[ IC50 is the half inhibitory concentration, which is required for MPs to inhibit 50% of liver cells after a specific exposure time. By calculating the cellular inhibition rate for successive administrations of different concentrations , a dose-response curve can be obtained to estimate the IC50 value.The smaller the IC50, the more potent the drug is.]

Comments 23:[329-which concentrations were used?]

Response 23:[ Thank you very much for your advice. We apologize for not describing it very clearly in 4.2.3. In fact the nanosized selenium concentrations we use are described at the beginning of this section and are 0.5, 1, 2, 2.5, and 5 umol/L concentrations. To avoid confusion, we have deleted the last sentence of 4.2.3.]

Comments 24:[343-The term "expression" is typically used in the context of gene expression. For biochemical indices, it would be more appropriate to use "levels" or "concentrations."]

Response 24:[ Thank you very much for your suggestion. We have changed “expression” to “levels” in the manuscript.]

Comments 25:[351-352-It is unclear what "BMC content" refers to]

Response 25:[ Thank you very much for your question, and we apologize for not explaining more about BMC, which stands for bone mineral content. we have added it to the manuscript!]

Comments 26:[373-374-Quantitative data?]

Response 26:[ Thank you very much for your question. We apologize for not being able to provide quantitative data on the H.E stained sections of the liver. In fact, in general, we determine the inflammatory pathologic features of the liver subjectively by ocular observation, including inflammatory cell infiltration and hepatocyte shape changes, which are also more difficult to quantify.]

Comments 27:[ 409-410-the concentration of SeNPs used?]

Response 27:[ Thank you very much for your advice. In fact, we mentioned in subsection 2.5 the concentration of selenium nanoparticles used to configure the in vivo experiments, which was 200 mg/L. We have marked this place in yellow!]

Comments 28:[417-418-quantitative data for MDA?]

Response 28:[ Thank you very much for your question. Our quantitative MDA data is presented in Figure 6. We are detecting MDA levels in serum by using the MDA kit built in Nanjing Jiancheng Bioengineering Institute.]

Comments 29:[ 431-432-This conclusion is valid based on the presented data]

Response 29:[Thank you very much for your advice. We also think that, by the data presented, nanosized selenium can reduce liver damage caused by microplastics]

Comments 30:[449-451-Including specific gene names would clarify which mitochondrial processes are being impacted.]

Response 30:[ Thank you very much for your suggestions, we have added the genes that specifically affect mitochondrial processes to the manuscript]

Comments 31:[ Discuss specific mechanisms in more depth.]

Response 31:[ Thank you very much for your advice. We have considered this, but in reality, we do not experiment with genes and proteins on cells. So there is no way for us to go more in-depth to discuss the deep-rooted mechanisms, and our paper mainly takes hepatocyte injury as a starting point to investigate the combined effects of nanosized selenium on attenuating microplastic-induced liver injury. However, your comments are very valuable, and our next study is preparing to focus on cellular experiments to investigate the deeper mechanisms.]

[1]Cunningham EM,Sigwart JD. Environmentally Accurate Microplastic Levels and Their Absence from Exposure Studies. Integr Comp Biol. 2019;59 (6):1485-1496.

[2]Yin K,Wang D,Zhang Y, et al. Polystyrene microplastics promote liver inflammation by inducing the formation of macrophages extracellular traps. J Hazard Mater. 2023;452:131236.

[3]Shi C,Han X,Guo W, et al. Disturbed Gut-Liver axis indicating oral exposure to polystyrene microplastic potentially increases the risk of insulin resistance. Environ Int. 2022;164:107273.

[4]K C, P.B,Maharjan, A.,Acharya, M.,Lee, D.,Kusma, S.,Gautam, R.,Kwon, J.T,Kim, C.,Kim, K.,Kim, H., & Heo, Y. (2023). Polytetrafluorethylene microplastic particles mediated oxidative stress, inflammation, and intracellular signaling pathway alteration in human derived cell lines. The Science of the total environment, 897, 165295.

[5]Danopoulos E, Twiddy M, West R, Rotchell JM. A rapid review and meta-regression analyses of the toxicological impacts of microplastic exposure in human cells. J Hazard Mater. 2022 Apr 5;427:127861.

[6]Cunningham EM,Sigwart JD. Environmentally Accurate Microplastic Levels and Their Absence from Exposure Studies. Integr Comp Biol. 2019;59 (6):1485-1496. doi:10.1093/icb/icz068

[7]Yin K,Wang D,Zhang Y, et al. Polystyrene microplastics promote liver inflammation by inducing the formation of macrophages extracellular traps. J Hazard Mater. 2023;452:131236. doi:10.1016/j.jhazmat.2023.131236

[8]Shi C,Han X,Guo W, et al. Disturbed Gut-Liver axis indicating oral exposure to polystyrene microplastic potentially increases the risk of insulin resistance. Environ Int. 2022;164:107273. doi:10.1016/j.envint.2022.107273

Reviewer 2 Report (Previous Reviewer 2)

Comments and Suggestions for Authors

The resubmitted manuscript is improved with better/improved description of materials and methods, introduction and discussion. No further revisions are required.

Author Response

Thank you very much

Round 2

Reviewer 1 Report (Previous Reviewer 4)

Comments and Suggestions for Authors

I have no more comments.

This manuscript is a resubmission of an earlier submission. The following is a list of the peer review reports and author responses from that submission.

Round 1

Reviewer 1 Report

Comments and Suggestions for Authors

The research is interesting regarding the wide use of microplastics. The study is well designed although some improvements or clarifications are necessary:

1. There is no need to write "degrees Celsius" or "revolutions per minute". Instead use the abbreviations: °C, rpm

2. The authors should describe the culture conditions of mouse hepatocytes to allow reproducibility of the experiment.

3. How did the authors choose the doses of PS-MPs? What is the IC50?

4. How were the doses for selenium chosen?

5. Why were ROS measured in cell cultures and not in tissue homogenates?

6. Why were 30-day old immature male mice used in the experiment instead of mature?

7. Section 2.5.2.1. should be revised. It is unclear how the tissues were processed prior to paraffin embedding and how was the H-E staining performed.

8. The authors should explain why did they place blood samples on water bath?

9.  Section 2.5.2.6. should be rewritten. Types of antibodies, buffers, dilutions, etc. should be given. How were the samples protected from light and drying? Were they kept in wet dark chamber?

10. Paraffin sections cannot be directly incubated with primary antibody. They must be deparaffinized first. 

11. It is unclear where were antioxidant indicators measured?

12. What is subspherical shape?

13. Lines 297-299. Why do authors write that there was no proliferative effect if in lines 293-294 they say that 2 micromol/L significantly stimulated proliferation?

14. Photograph quality must be improved. It is difficult to see the changes the authors describe. I suggest the authors to show light microscopy and TEM images in different panels. 

15. Why didn't the authors use the in vivo model for all tests? What advantages give the in vitro tests?

Comments on the Quality of English Language

Good quality English. There are some minor technical and grammar mistakes. I recommend the use of "mouse liver" and "mouse hepatocyte cell suspension" instead of "mice liver" and "mice hepatocyte..."

Author Response

Thank you very much for your comments, the changes to your question are highlighted in red in the manuscript

Comments 1:[There is no need to write "degrees Celsius" or "revolutions per minute". Instead use the abbreviations: °C, rpm]

Response 1:[This has been corrected]

Comments 2:[The authors should describe the culture conditions of mouse hepatocytes to allow reproducibility of the experiment.]

Response 2:[This has been added]

Comments 3: [How did the authors choose the doses of PS-MPs? What is the IC50?]

Response 3:[Concentrations of microplastics vary in various regions or environmental areas around the globe, and rather than modelling the concentration of microplastics in a particular environment in reality, our study focuses on exploring the definitive effects of microplastics on the tissues of living organisms. At the same time, studies have shown that human plastic output increased exponentially between 2004-2014, so the dose we chose is somewhat predictive[1]. Although 1mg/kg is a larger than common environmental dose, there are still many studies that use this dose to explore the effects of MPs on various tissues and organs of organisms[2  ,3]. In our pre-experiment, we found that a concentration of 1mg/kg of MPs is less prone to agglutination and precipitation, which ensures that the amount of microplastics ingested by each mouse is close to the same amount.]

 We apologise for not explaining the meaning of IC50, which is the semi-inhibitory concentration of the measured substance.

Comments 4: [How were the doses for selenium chosen?]

Response 4:[Our laboratory has explored the therapeutic dosage of nanosized selenium in the range of 0.5 mg/kg,Nano selenium at 1mg/kg has therapeutic effect on cardiomyocytes and heart[5].]

Comments 5:[Why were ROS measured in cell cultures and not in tissue homogenates?]

Response 5:[Thank you very much for your question, in fact, our research on the effects of MPs on the liver consists of two parts, one of which has been published in Ecotoxicology and Environmental Safety[4], and that paper has detailed data on ROS in the liver, and the present paper focuses mainly on the changes in hepatic inflammation induced by MPs and the effects on hepatocytes.]

Comments 6:[Why were 30-day old immature male mice used in the experiment instead of mature?]

Response 6:[This issue is invaluable. Immature mice are more responsive to exogenous substances, and it is also important to explore whether there are any effects of MPs on the development of mice, as indicated by bone density, body weight, etc.]

Comments 7:[Section 2.5.2.1. should be revised. It is unclear how the tissues were processed prior to paraffin embedding and how was the H-E staining performed.]

Response 7:[This has been added]

Comments 8: [The authors should explain why did they place blood samples on water bath?]

Response 8:[Blood samples are more likely to produce serum at 37°C]

Comments 9:[Section 2.5.2.6. should be rewritten. Types of antibodies, buffers, dilutions, etc. should be given. How were the samples protected from light and drying? Were they kept in wet dark chamber?]

Response 9: [A view we couldn't agree with more and have added this to the manuscript as per your request]

Comments 10:[Paraffin sections cannot be directly incubated with primary antibody. They must be deparaffinized first.]

Response 10:[Thank you for your valuable comments, we have corrected them!]

Comments 11: [It is unclear where were antioxidant indicators measured?]

Response 11:[I'm sorry we didn't make this clear, we measured antioxidant indicators in mouse serum. We have added this to the manuscript.]

Comments 12:[What is subspherical shape?]

Response 12:[Thank you very much for identifying our error in detail, what we were actually trying to convey was approximate spherical shape. This has been corrected]

Comments13.:Lines 297-299. Why do authors write that there was no proliferative effect if in lines 293-294 they say that 2 micromol/L significantly stimulated proliferation?

Response 13:[Thank you very much for pointing out this misrepresentation for us, we have removed the contradictory part. What we actually wanted to express was that low-dose nanosized selenium (0.5 umol/L) had no effect of stimulating cell proliferation, whereas high-dose nanosized selenium had an effect of stimulating cell proliferation.]

Comments 14:[Photograph quality must be improved. It is difficult to see the changes the authors describe. I suggest the authors to show light microscopy and TEM images in different panels.]

Response 14:[Your comments are invaluable to our manuscript, and we have combined a variety of images due to typographical and article flow considerations. However, we will eventually upload the highest resolution version of a single image to our editors for easy viewing by our readers.]

Comments 15:[Why didn't the authors use the in vivo model for all tests? What advantages give the in vitro tests?]

Response 15: [In the in vitro model, we focus on the proliferative effects of MPs and SeNPs on cells, while in the in vivo experiments, we focus on the rna transcription and protein expression components.]

language:[We have removed all "mice liver" and "mice hepatocyte..." have been changed to  "mouse liver" and "mouse hepatocyte cell suspension" ]

[1]Cunningham EM,Sigwart JD. Environmentally Accurate Microplastic Levels and Their Absence from Exposure Studies. Integr Comp Biol. 2019;59 (6):1485-1496. doi:10.1093/icb/icz068

[2]Yin K,Wang D,Zhang Y, et al. Polystyrene microplastics promote liver inflammation by inducing the formation of macrophages extracellular traps. J Hazard Mater. 2023;452:131236. doi:10.1016/j.jhazmat.2023.131236

[3]Shi C,Han X,Guo W, et al. Disturbed Gut-Liver axis indicating oral exposure to polystyrene microplastic potentially increases the risk of insulin resistance. Environ Int. 2022;164:107273. doi:10.1016/j.envint.2022.107273

[4]Shen Q,Liu YJ,Qiu TT, et al. Microplastic-induced NAFLD: Hepatoprotective effects of nanosized selenium. Ecotoxicol Environ Saf. 2024;272:115850. doi:10.1016/j.ecoenv.2023.115850

[5]Yang X,Fu Y,Liu J, et al. A new application of nano-selenium: rescue of CK2 and mitochondria from oxidative stress to prevent cardiac hypertrophy. Nanomedicine (Lond). 2023;18 (21):1421-1439. doi:10.2217/nnm-2022-0325

Reviewer 2 Report

Comments and Suggestions for Authors

This paper by Qi Shen and colleagues deals with a very important and very current topic, i.e. microplastic (PS-MPs) pollution. The authors focused on nano-selenium (SeNPs) and its role in protecting the liver/ameliorating the liver damage induced by PS-MPs. This paper is interesting and adds data to study this serious problem, seeks to understand the effects on human/public health and allow the study of possible solutions. 

Major points to address:

- The methods/protocol followed for in vitro experiments are well described, while in vivo methods lack information. In section "2.5.2. Animal experiment grouping" they do not state the dosage of PS-MPs used to treat the animals. Additionally, they refer to mice "exposed to microplastic" or "contaminated", but they never described how they administered or challenged mice with PS-MPs nor describe the dosages used.

- Fig 3 and fig 4: what is the meaning of "L-PS-MPs, M-PS-MPs and H-PS-MPs"? Are these the dosages of microparticles used? what concentration are they referring to? 

- In the results section "4.2.2. Detection of PS-MPs on hepatocytes by CCK8" they describe the in vitro experiment with different dosage of PS-MPs. Are the doses tested realistic? what is the concentration found in polluted area?

- In the section "4.2.3. Detection of SeNPs on hepatocytes by CCK8" they used selenium (SeNPs) at the concentration 0.5-2.5 μmol/L. Are these concentrations far or close to toxic concentration for mice/humans?

- In the section "Animal Testing I and II" they described a liver damage and inflammation, caused by what? Did they find microparticles within the liver of treated mice? did they check for liver fibrosis? they should perform also a MPO staining to evaluate the level of myeloid cells activation.

- In "4.3.1.2. Other effects of PS-MPs on mice" they checked bones and fat content in the tissues. What tissues? Did they check other tissues, i.e. spleen, kidney, lung, heart, intestine? Did they find PS-MPs in these organs? Where these other organs affected by PS-MPs?

- Lines 378-379 they wrote "Furthermore, there was a notable rise in the fat content within the mouse tissues in the PS-MPs group compared to the control group". How do they explain this increase in fat content induced by PS-MPs?

- SeNPs was used at a dosage of 1 mg/kg/day. Wouldn't this dosage be toxic for human beings?

- In vitro studies: they describe a proliferative effect on hepatocytes by SeNPs and a toxic effect my PS-MPs. Did they check on other cell types, i.e. endothelial cells and/or macrophages, at the same concentration? Liver sinusoidal endothelial cells and Kupffer cells are the first cell forming a barrier diving hepatocytes from the bloodstream.

- How do they think selenium could be used and administered in humans? Wouldn't it be enough to increase that in the diet?

Minor points:

- line 78 "at 4 degrees Celsius" is repeated

- lines 274-275: "0-24h cells were in the stagnant phase, 24h-48h cells were in the logarithmic phase and 48h-72h cells were in the equilibrium phase." please change "stagnant phase" with "lag phase" and "equilibrium phase" with "stationary phase".

- line 291: nano is repeated "nano nano-selenium"

- line 310: "Serum biochemical parameters were measured in PS-MPs-induced liver tissue". Did they mean "Serum biochemical parameters were measured in plasma/serum/blood of PS-MPs-challenged mice"?

Comments on the Quality of English Language

Please improve the Materials and methods section and check for typos throughout the text

Author Response

Comments 1:[The methods/protocol followed for in vitro experiments are well described, while in vivo methods lack information. In section "2.5.2. Animal experiment grouping" they do not state the dosage of PS-MPs used to treat the animals. Additionally, they refer to mice "exposed to microplastic" or "contaminated", but they never described how they administered or challenged mice with PS-MPs nor describe the dosages used.]

Response 1:[Thank you very much for your careful review, we are using gavage to let mice ingest MPs and SeNPs, and the dose of MPs is 1mg/kg/d.The content has now been added to the manuscript]

Comments 2:[Fig 3 and fig 4: what is the meaning of "L-PS-MPs, M-PS-MPs and H-PS-MPs"? Are these the dosages of microparticles used? what concentration are they referring to?]

Response 2:[We apologise for not providing a detailed explanation of the doses of the three groups of MPs, which only appeared in 2.5.1." L-PS-MPs, M-PS-MPs and H-PS-MPs" refer to the three groups of mice exposed to low(PS-MPs 0.1mg/kg/d), medium(PS-MPs 1mg/kg/d) and high(PS-MPs 10mg/kg/d) concentrations of MPs, respectively.] 

Comments 3:[In the results section "4.2.2. Detection of PS-MPs on hepatocytes by CCK8" they describe the in vitro experiment with different dosage of PS-MPs. Are the doses tested realistic? what is the concentration found in polluted area?]

Response 3:[In the results section "4.2.2. Detection of PS-MPs on hepatocytes by CCK8" they describe the in vitro experiment with different dosage of PS-MPs. Are the doses tested realistic? what is the concentration found in polluted area?]

Comments 4:[In the results section "4.2.2. Detection of PS-MPs on hepatocytes by CCK8" they describe the in vitro experiment with different dosage of PS-MPs. Are the doses tested realistic? what is the concentration found in polluted area?]

Response 4:[The lowest concentration we set (50ug/ml) is close to the reality of high microplastic concentrations. And the higher concentration of microplastic suspension was set to explore the concentration limit and the degree of significant damage to the cells]

Comments 5:[In the section "4.2.3. Detection of SeNPs on hepatocytes by CCK8" they used selenium (SeNPs) at the concentration 0.5-2.5 μmol/L. Are these concentrations far or close to toxic concentration for mice/humans?]

Response 5:[Normal adult daily intake of selenium in dietary fibre is 1ug/1kg/d. This is close to the nano- selenium concentration we set]

Comments 6:[In the section "Animal Testing I and II" they described a liver damage and inflammation, caused by what? Did they find microparticles within the liver of treated mice? did they check for liver fibrosis? they should perform also a MPO staining to evaluate the level of myeloid cells activation]

Response 6:[Your question is invaluable, and we suspect that the inflammation and damage to these liver cells would be caused by microplastics. Since we did not fluorescently stain the microplastics, we cannot prove whether the microplastics entered the mouse liver. However, a large number of previous studies (including fluorescent staining experiments on microplastics) have demonstrated that microplastics smaller than 50 um enter and accumulate in the liver. The main focus of our experiment was to investigate the inflammatory changes in hepatocytes induced by microplastics, while liver fibrosis is the result of long-term inflammation in the liver, which we may investigate in future experiments.]

Comments 7:[In "4.3.1.2. Other effects of PS-MPs on mice" they checked bones and fat content in the tissues. What tissues? Did they check other tissues, i.e. spleen, kidney, lung, heart, intestine? Did they find PS-MPs in these organs? Where these other organs affected by PS-MPs?]

Response 7:[We are sorry that we did not explain this in detail in the content, in fact, using the bone densitometer can be detected by X-ray of the density changes of the components of the body of mice, in the ratio can be obtained by the results of the indicators such as bone density and fat ratio. At the same time, we did not design the experiment to detect the presence of microplastics in bone and fat tissues, and your question reminds me that we may be able to conduct this type of research in the future.]

Comments 8:[SeNPs was used at a dosage of 1 mg/kg/day. Wouldn't this dosage be toxic for human beings?]

Response 8:[Your question is very valid, our lab has previously investigated the therapeutic concentration range of nano selenium in chicken cardiomyocytes and chickens, as well as the toxicity range of nano selenium in mice, and the results have proven that there is no toxicity associated with the consumption of biogenic nano selenium at 1mg/kg/d!]

Comments 9:[In vitro studies: they describe a proliferative effect on hepatocytes by SeNPs and a toxic effect my PS-MPs. Did they check on other cell types, i.e. endothelial cells and/or macrophages, at the same concentration? Liver sinusoidal endothelial cells and Kupffer cells are the first cell forming a barrier diving hepatocytes from the bloodstream.]

Response 9:[Your suggestions are all very innovative and inspiring. We have not been able to take this into account and will explore this mechanism in subsequent experiments!]

Comments 10:[How do they think selenium could be used and administered in humans? Wouldn't it be enough to increase that in the diet?]

Respones 10:[We are very happy to discuss this topic with you. Firstly, selenium in dietary fibre is sufficient for humans to maintain normal physiology. However, selenium bioavailability and quantity are deficient in humans who have been subjected to chronic exogenous stimuli and whose livers are in a pathological state. Selenoproteins are important components of antioxidant and anti-inflammatory responses within organisms, and selenium, as the main substance for selenoprotein synthesis, is necessary to ensure its adequate intake in response to excessive depletion. In food, selenium is found in organic forms such as selenomethionine and selenocysteine, which need to be metabolised and utilised in the organism, and in selenium-deficient areas there is a need for safer and more highly utilised nano-selenium supplements.]

Comments 11:[line 78 "at 4 degrees Celsius" is repeated]

Response 11:[Thank you for the reminder, the duplicate capacity has been removed.]

Comments 12:[lines 274-275: "0-24h cells were in the stagnant phase, 24h-48h cells were in the logarithmic phase and 48h-72h cells were in the equilibrium phase." please change "stagnant phase" with "lag phase" and "equilibrium phase" with "stationary phase".]

Response 12:[Thank you for the kind reminder, we have corrected it!]

Comments 13:[line 291: nano is repeated "nano nano-selenium"]

Response 13:[We have removed it]

Comments 14:[line 310: "Serum biochemical parameters were measured in PS-MPs-induced liver tissue". Did they mean "Serum biochemical parameters were measured in plasma/serum/blood of PS-MPs-challenged mice"?]

Response 14:[We have corrected this, and we used mouse serum]

Language:[Please improve the Materials and methods section and check for typos throughout the text]

Response:[Thanks for the suggestion, we'll touch up the material methods section next!]

Reviewer 3 Report

Comments and Suggestions for Authors

The manuscript of Qi Shen et al. addressed an interesting topic regarding the potential therapeutic effect of nano-selenium particles (SeNPs) against microplastics (PS-MPs)-induced liver injury. The authors utilized a combination of in vitro and in vivo methods to demonstrate the protective effect of SeNPs against hepatocyte damage and liver injury caused by acute and chronic exposure to PS-MPs. The study's mechanistic analysis revealed that the protective effect of SeNPs may be attributed to the modulation of pro-inflammatory pathways, such as NF-KB/NRLP3, and the regulation of mitochondrial dynamics.

While the findings of the study are both novel and well supported by an experimental methodology that combines in vitro and in vivo approaches, certain aspects of the research are not fully clear to the reviewer. On the other hand, the authors must thoroughly revise several sections of the manuscript, particularly the Materials and methods and Discussion. Therefore, further, and extensive revision and improvement of the manuscript are needed to meet the quality standards for its publication on Nutrients.

Some of these suggestions included:

·       Lines 39-43 should be rephrased, as redundant information is provided in that paragraph.

·       The inclusion of appropriate references in lines 44-47 to support the information provided by the authors is highly recommended.

·       To ensure clarity and coherence, it is suggested that the toxicological mechanisms of action of PS-MPs, such as oxidative stress, mitochondrial impairment, inflammatory pathways, and other related concepts, be incorporated in the Introduction section. This will improve the manuscript's clarity and coherence.

·       The concluding sentence of the Introduction section (i.e., lines 61-63) ought to include information about the in vitro model utilized by the authors to investigate the consequences of SeNPs on PS-MPs liver-related damage.

·       The materials and methods section should be extensively revised, and several of the main text elements within the subheading should be rephrased. In this regard, although the authors have provided a comprehensive description of the methods employed in the manuscript, potential readers may face challenges in replicating the experiments. It is crucial to address the following questions in this section:

·       In the manuscript, it is essential to specify which type of microparticles (MPs) the authors utilized in their research. Were polystyrene (PS) MPs employed? This detail must be clearly stated in the text.

·       Furthermore, the duration for which the PS-MPs and SeNPs were maintained in solution ought to be specified. Additionally, the method of storage for these particles should be disclosed.

·       Please ensure that the authors provide complete information on all reagents and instruments used in the study, including the brand or product catalog number. This information should be systematically indicated in the Materials and Methods section to avoid any confusion.

·       The authors should add an additional subheading in the in vitro experimental design section to provide information on the cell culture conditions of the mice's normal hepatocytes. This information is crucial for readers to understand the experimental setup. Additionally, the authors should include details on the cell culture media composition used in the manuscript.

·       To ensure transparency in the study, the authors should clearly indicate the cell culture media composition used for the in vitro exposure to the PS-MPs. Specifically, the authors should specify whether the cell culture media used contained FBS or not, as this information can significantly impact the results and the conclusions drawn from the study (i.e., corona effect).

·       To enhance the readability of the subheading cell growth curve determination, the authors should revise the main text to make it more comprehensible to potential readers. The term "resuscitate" should be avoided.

·       To provide context for the in vitro testing described in the subheading cell growth curve determination, the authors should explain the rationale behind this experiment and its significance to the research.

·       To improve the clarity of the subheadings 2.4.2. and 2.4.3., the authors should revise the titles to more accurately reflect the content of the subheadings. Specifically, the authors should indicate that they are detecting the cytotoxic profile of PS-MPs and SeNPs, rather than their presence in the in vitro system. These revisions should also be applied to the subheadings 4.2.2. and 4.2.3. in the Results section.

·       The absorbance at which the CCK8 was detected should be indicated in the manuscript as well as how the CCK8 data were analysed.

·       The authors should specify how the ROS detection was analyzed, including whether it was normalized against cellular numbers.

·       In the subheading 2.5.2., the authors should clearly state the frequency of animal exposure to seNPs and PS-MPs.

·       The authors should indicate in which tissue or biological fluid the antioxidant indicators were measured in the in vivo model.

·       The current section 2.5.2.6 necessitates substantial revision. Numerous essential details have been overlooked, such as (i) the composition of buffers, (ii) the method used to deparaffinize samples, (iii) the thickness of the sections, (iv) the antigen retrieval process, (v) the dilutions and references for both primary and secondary antibodies, and (vi) the exposure times and temperatures for the primary and secondary antibodies.

·       The current section 2.5.2.7 necessitates substantial revision. Numerous essential details have been overlooked, such as (i) the cutoff value for RNA absorbance ratio, (ii) cDNA reference kits, (iii) specifications of the PCR instruments, (iv) chemicals used for qPCR reactions (e.g. SYBR Green), (v) genes and sequences of primers, (vi) the housekeeping gene selected and evidence of its gene expression stability, and (vii) the methodology employed for analyzing gene expression results (e.g. delta delta ct method).

·       The current section 2.5.2.8 necessitates substantial revision. Numerous essential details have been overlooked, including (i) the composition of buffers, (ii) the method of sample separation during electrophoresis and the percentage of SDS-PAGE gel, (iii) the dilutions and references for both primary and secondary antibodies, (iv) the exposure times and temperatures for the primary and secondary antibodies, and (v) the technique utilized for analyzing protein expression results.

·       The reviewer is uncertain about the statistical analysis that the authors have conducted. Specifically, the reviewer is wondering why the authors chose to use a t-test rather than another statistical parametric test such as ANOVA or even statistical non-parametric testing. Additionally, authors should clarify whether they tested the normality of their data, as this would determine whether a parametric test is appropriate. Furthermore, authors should provide additional information on the sample size, including the number of biological and technical replicates, as well as how the data are expressed (e.g. median ± SD) in the different in vitro and in vivo experiments of the manuscript.

·       The authors produced SeNPs using a bacterial-based system. The reviewer wonders how the authors control the potential contamination of bacterial products in the SeNPs upon their production.

·       The authors utilized a broad range of PS-MP concentrations in their in vitro studies. The reviewer questions the biological significance of these concentrations. Are they relevant? Are they supported by previous in vitro studies? This also applies to the PS-MP concentrations used in the in vivo studies. Are these concentrations relevant? Are there other in vivo studies that have employed similar concentrations?

·       In line with the previous comment, are the SeNP concentrations used for both in vitro and in vivo experiments biologically relevant? Are they based on previous reports?

·       The reviewer notes that the time points 4 and 12 hours are not included in Figure 2, panel A, in the Results section, subheading 4.1 despite that these time points are indicated to be evaluated in the Materials and Methods section.

·       Considering the red coloration of the SeNPs, the reviewer questions whether the authors have evaluated the potential impact of SeNPs on both the CCK8 and ROS readouts. This is crucial to avoid potential technical artifacts that could affect the results obtained.

·       The concentrations of SeNPs that were evaluated for their anti-antioxidant effect should be indicated in the main text of the manuscript (i.e. line 297) as well as in the Figure 2 caption.

·       The reviewer also wants to know how the fluorescent signal related to the ROS quantification was processed by the authors and if this signal was normalized considering the number of cells. Additionally, it would be beneficial for the manuscript if the authors could include not only a representative picture of the in vitro ROS-related experiments but also a quantitative quantification of these experiments.

·       Information related to Figure 3, panels D, E, and F should be incorporated in the main text of the manuscript.

·       The subheading title 4.3.1.2. should be rephrased into a more specific title that summarizes the findings indicated in the main text of the subheading.

·       The reviewer wants to know the biological relevance of evaluating the BMC and BMD upon exposure to the PS-MPs. Is there any indication that the PS-MPs can affect these read-outs?

·       The authors should clearly indicate why the highest dose of PS-MPs was selected for subsequent in vivo experiments. The sentence indicated in lines 333 and 334 is vague in this regard.

·       In Figure 4, panel A, the reviewer cannot discern the condition depicted in the image. To provide clarity, a more detailed description should be included in the figure caption.

·       In subheading 4.3.2.1, the authors characterize their findings as 'significant' (i.e., line 345). However, they present only a representative liver histology image in Figure 5. Does the authors' analysis include quantitative data that supports their conclusions? If so, this quantitative data should be included in Figure 5.

·       The reviewer would like to know how the authors quantified the fat tissue content depicted in Figure 6, panel H, as well as in line 379 of the main text.

·       The reviewer is curious as to why the authors referred to the NRLP3 inflammasome pathway as an inflammatory vesicle pathway.

·       The reviewer is uncertain about the number of animals included in the western blot quantification depicted in Figure 7, panels C-I.

·       To enhance the manuscript, the authors should place the subheading 4.3.2.5 before 4.3.2.4. This arrangement would allow the authors to initially present the transcriptional effects of the treatments and subsequently describe their translational effects.

·       The content in line 413 requires correction, as several of the targets mentioned by the authors are translational modifications that cannot be observed at the gene expression level (e.g., cleaved caspase 1 or phosphorylated NF-KB).

·       It would benefit the manuscript if the subheading 4.3.2.6 provided more information about the biological function of the selected mitochondrial genes in the main text.

·       In the Discussion portion of the manuscript, several paragraphs require revision, shortening, and clarification to ensure that the focus is primarily on comparing the obtained results with previous reports, rather than providing general information.

·       Findings related to the BMC and BMD readouts as well as the total content of fact upon exposure to the PS-MPs are not discussed.

·       The reviewer has questioned why additional in vitro experiments were not conducted to evaluate the potential impact of the PS-MPs on the pro-inflammatory and mitochondrial pathways.

Comments on the Quality of English Language

·       The manuscript presents several issues that require attention, including typographical errors (e.g. "caspase1" in line 400, etc.), inconsistencies (e.g. 4 weeks vs 28 days, numbers written in numerical characters vs full letters, etc.), incorrect abbreviations, scientific notation (e.g. centrifuge speed, cell densities, etc.), and non-scientific language (e.g. "revived cells," "animals were contaminated," etc.). It is recommended that the authors carefully review and revise the manuscript to address these concerns.

·       The names of the genes should be revised and written according to the scientific guidelines.

·       The font and size of the Figure 5 caption should be modified and uniformized with the rest of the figure’s captions of the manuscript.

·       The figure captions should also be revised to include the sample size, including biological and technical replicates, the statistical test used for comparisons, and references to the scale bar in all captions.

·       The tense of the Materials and Methods section should be changed to past tense.

·      In general, it would be beneficial for the manuscript if the English style were reviewed and revised to improve the overall clarity of the document.

Author Response

Thank you very much for your valuable comments on this manuscript, I will highlight the changes in yellow!

Comments 1:[Lines 39-43 should be rephrased, as redundant information is provided in that paragraph.]

Response 1:[ Thank you very much for your suggestion, it has been changed to “Microplastic particles have been found in several organs of living organisms while posing serious health risk, while accumulation in the liver of mice has been widely reporte.”]

Comments 2:[The inclusion of appropriate references in lines 44-47 to support the information provided by the authors is highly recommended.]

Response 2:[ Thank you for your advice. We have added more references here.]

Comments 3:[To ensure clarity and coherence, it is suggested that the toxicological mechanisms of action of PS-MPs, such as oxidative stress, mitochondrial impairment, inflammatory pathways, and other related concepts, be incorporated in the Introduction section. This will improve the manuscript's clarity and coherence.]

Response 3:[ Thank you for your advice. Reports related to the mechanism of MPs-induced liver injury have been added to the Introduction]

Comments 4:[ The concluding sentence of the Introduction section (i.e., lines 61-63) ought to include information about the in vitro model utilized by the authors to investigate the consequences of SeNPs on PS-MPs liver-related damage.]

Response 4:[ Thank you very much for your advice. We should indeed include the experimental design of microplastics and nanoselenium on cells in the summary of the introduction. We have added the relevant content here.]

Comments 5:[ In the manuscript, it is essential to specify which type of microparticles (MPs) the authors utilized in their research. Were polystyrene (PS) MPs employed? This detail must be clearly stated in the text.]

Response 5:[ Thank you for your advice. The chemistry of the PS-MPs we use is polystyrene. We have added this information to the manuscript]

Comments 6:[ Furthermore, the duration for which the PS-MPs and SeNPs were maintained in solution ought to be specified. Additionally, the method of storage for these particles should be disclosed.]

Response 6:[ Thank you for your advice. Preservation methods for MPs and SeNPs have been published in the manuscript]

Comments 7:[Please ensure that the authors provide complete information on all reagents and instruments used in the study, including the brand or product catalog number. This information should be systematically indicated in the Materials and Methods section to avoid any confusion.]

Response 7:[Thank you very much for your suggestions. We have endeavoured to add as much information as possible about the instruments used to the manuscript.]

Comments 8:[ The authors should add an additional subheading in the in vitro experimental design section to provide information on the cell culture conditions of the mice's normal hepatocytes. This information is crucial for readers to understand the experimental setup. Additionally, the authors should include details on the cell culture media composition used in the manuscript.]

Response 8:[ Thank you very much for your advice. We've added a new section 2.4.1 to describe hepatocyte culture methods!]

Comments 9:[ To ensure transparency in the study, the authors should clearly indicate the cell culture media composition used for the in vitro exposure to the PS-MPs. Specifically, the authors should specify whether the cell culture media used contained FBS or not, as this information can significantly impact the results and the conclusions drawn from the study (i.e., corona effect).]

Response 9:[ Thank you very much for your advice. We have provided the specific composition of the cell culture medium in the Material Methods. At the same time we have not added FBS to the culture medium.]

Comments 10:[To enhance the readability of the subheading cell growth curve determination, the authors should revise the main text to make it more comprehensible to potential readers. The term "resuscitate" should be avoided.]

Response 10:[ Thank you for your suggestion, the section 2.4.2 has been modified in its entirety!]

Comments 11:[To provide context for the in vitro testing described in the subheading cell growth curve determination, the authors should explain the rationale behind this experiment and its significance to the research.]

Response 11:[ Thank you for your advice. We have added the context of this experiment to the material methods.To ensure that cells exposed to microplastics and nanosized selenium were in equilibrium growth phase and normal, we measured the growth curves of hepatocytes.]

Comments 12:[ To improve the clarity of the subheadings 2.4.2. and 2.4.3., the authors should revise the titles to more accurately reflect the content of the subheadings. Specifically, the authors should indicate that they are detecting the cytotoxic profile of PS-MPs and SeNPs, rather than their presence in the in vitro system. These revisions should also be applied to the subheadings 4.2.2. and 4.2.3. in the Results section.]

Response 12:[ Thank you for your advice. The title of 2.4.2 has been changed to "Effect of PS-MPs on hepatocyte proliferation". Changed the title of 2.4.3 to "Effect of SeNPs on hepatocyte proliferation".]

Comments 13:[The absorbance at which the CCK8 was detected should be indicated in the manuscript as well as how the CCK8 data were analysed.]

Response 13:[ Thank you for your advice. We have added the instruments and conditions for detecting absorbance to the manuscript]

Comments 14:[The authors should specify how the ROS detection was analyzed, including whether it was normalized against cellular numbers.]

Response 14:[ Thank you very much for your question. We have added an analysis of how ROS was analysed to the manuscript. In the meantime we made the cell concentration per well as consistent as possible before going for the ros assay.]

Comments 15:[In the subheading 2.5.2., the authors should clearly state the frequency of animal exposure to seNPs and PS-MPs.]

Response 15:[ Thank you very much for your advice. We have revised subsection 2.5 with the addition of the exposure frequencies of MPs and SeNPs]

Comments 16:[ The authors should indicate in which tissue or biological fluid the antioxidant indicators were measured in the in vivo model.]

Response 16:[ We apologise for any inconvenience caused by our negligence. The antioxidant indexes in mouse serum that we assayed have been added to the manuscript in the meantime.]

Comments 17:[ The current section 2.5.2.6 necessitates substantial revision. Numerous essential details have been overlooked, such as (i) the composition of buffers, (ii) the method used to deparaffinize samples, (iii) the thickness of the sections, (iv) the antigen retrieval process, (v) the dilutions and references for both primary and secondary antibodies, and (vi) the exposure times and temperatures for the primary and secondary antibodies.]

Response 17:[ Thank you for your advice. The immunofluorescence staining process has been modified in its entirety and the above has been added]

Comments 18:[The current section 2.5.2.7 necessitates substantial revision. Numerous essential details have been overlooked, such as (i) the cutoff value for RNA absorbance ratio, (ii) cDNA reference kits, (iii) specifications of the PCR instruments, (iv) chemicals used for qPCR reactions (e.g. SYBR Green), (v) genes and sequences of primers, (vi) the housekeeping gene selected and evidence of its gene expression stability, and (vii) the methodology employed for analyzing gene expression results (e.g. delta delta ct method).]

Response 18:[ Thank you very much for your suggestion. the experimental methods in the qt-qcr section are modified in order to follow the above requirements. In addition, we will also upload the gene sequences into the Supplementary Material.]

Comments 19:[ The current section 2.5.2.8 necessitates substantial revision. Numerous essential details have been overlooked, including (i) the composition of buffers, (ii) the method of sample separation during electrophoresis and the percentage of SDS-PAGE gel, (iii) the dilutions and references for both primary and secondary antibodies, (iv) the exposure times and temperatures for the primary and secondary antibodies, and (v) the technique utilized for analyzing protein expression results.]

Response 19:[ Thank you for your advice. We have revised and added to this section in accordance with the above requirements]

Comments 20:[ The reviewer is uncertain about the statistical analysis that the authors have conducted. Specifically, the reviewer is wondering why the authors chose to use a t-test rather than another statistical parametric test such as ANOVA or even statistical non-parametric testing. Additionally, authors should clarify whether they tested the normality of their data, as this would determine whether a parametric test is appropriate. Furthermore, authors should provide additional information on the sample size, including the number of biological and technical replicates, as well as how the data are expressed (e.g. median ± SD) in the different in vitro and in vivo experiments of the manuscript.]

Response 20:[ Thank you very much for your suggestion, which is very helpful for us to find out the errors in the paper and improve the paper. We would like to sincerely apologise for not presenting the correct statistical methods in the manuscript due to communication and coordination problems of the writers. In fact, for comparisons between multiple groups, we used the ANOVA method. It was only for individual single group to single group comparisons that t-tests were used, but this is not relevant to the content of this paper. We have revised the correct statistical methods into the manuscript.

Comments 21:[ The authors produced SeNPs using a bacterial-based system. The reviewer wonders how the authors control the potential contamination of bacterial products in the SeNPs upon their production.]

Response 21:[ Thank you very much for your question. We are more than happy to answer your questions. Firstly for the laboratory and personnel, we strictly follow the biosafety regulations for microbiological laboratories to ensure that there is no leakage of any microorganisms or infections during the production process. Of course the most important thing is how to control bacterial contamination of nanoselenium of bacterial origin. In the process of making nano selenium, we will separate the nano selenium particles from the bacteria and break the bacteria through ultra-high speed centrifugation, and grind them in liquid nitrogen to make the bacteria get further broken. Then wash the bacterial fragments in the precipitation by octanol. Finally, the obtained selenium nanoparticles are observed through an electron microscope, and no surviving bacteria appear in the field of view. At this point it can be assured that the powder is not contaminated with bacteria.

Comments 22:[ The authors utilized a broad range of PS-MP concentrations in their in vitro studies. The reviewer questions the biological significance of these concentrations. Are they relevant? Are they supported by previous in vitro studies? This also applies to the PS-MP concentrations used in the in vivo studies. Are these concentrations relevant? Are there other in vivo studies that have employed similar concentrations?]

Response 22:[ Microplastic concentrations vary in different regions or environmental areas around the globe, and rather than modelling realistic microplastic concentrations in specific environments, our study focuses on exploring the explicit effects of microplastics on organismal tissues. At the same time, studies have shown an exponential increase in the production of plastics in humans between 2004-2014, making our chosen dose somewhat predictive [1]. Although the environmental dose of 1 mg/kg is larger than common doses, there are still many studies using this dose to explore the effects of microplastics on various tissues and organs of organisms [2,3]. In our preliminary experiments, we found that MPs at a concentration of 1 mg/kg did not readily agglutinate and precipitate, which ensured that the amount of microplastics ingested by each mouse was close to the same amount. Also to ensure consistency between in vitro and in vivo studies, we intend to use different gradients of microplastic concentration to observe the cellular response to them. There have been studies where THP-1 cell lines were treated with microplastic suspensions ranging from 1-1000ug/ml to assess their toxicity to the cells[4]. It has also been shown that concentrations of 10 μg/mL (5–200 µm), had an adverse effect on cell viability, and 20 μg/mL (0.4 µm) on cytokine release.[5]

Comments 23:[  In line with the previous comment, are the SeNP concentrations used for both in vitro and in vivo experiments biologically relevant? Are they based on previous reports?]

Response 23:[ Thank you very much for your question. Based on our lab's previous research on nano-selenium, we found that 0.5-1mg/kg of nano-selenium was effective in alleviating cardiac hypertrophy in chickens[6]. There are also a number of studies reporting the therapeutic effects of different doses of nano-selenium. For example, the study used 5mg/kg of nanoselenium to alleviate heat stress-induced oxidative damage in organs by activating the nrf2 pathway[7]. One study used 0.5mg/kg of nanosized selenium to treat coccidia-induced oxidative stress in the jejunum of mice[8]. Since the specific mechanism of nanoselenium's action in the organism is not fully understood, we conservatively chose the nanoselenium concentration of 1 mg/kg, which has been frequently reported in the past, while cellular experiments with nanoselenium have rarely been reported, and based on the molecular mass of selenium, we chose the nanoselenium suspension with a concentration of 5 umol/L, which approximates to the dosage of 0.5 mg/kg.Considering the cellular fragility and the conservatism of the experiment, we chose to gradually reduce the concentration gradient to explore its therapeutic effect.

 Comments 24:[The reviewer notes that the time points 4 and 12 hours are not included in Figure 2, panel A, in the Results section, subheading 4.1 despite that these time points are indicated to be evaluated in the Materials and Methods section]

Response 24:[ Thank you very much for your question. Due to the fact that the change in the number of cells growing cells is not obvious enough at the 4th and 12th hour nodes, in order to ensure the beauty and consistency of the pictures. We chose to ignore this part on the picture.]

Comments 25:[Considering the red coloration of the SeNPs, the reviewer questions whether the authors have evaluated the potential impact of SeNPs on both the CCK8 and ROS readouts. This is crucial to avoid potential technical artifacts that could affect the results obtained]

Response 25:[ Thank you very much for your very insightful question. Firstly, the red colour shown by its selenium nano-suspension has been very faint at the concentration gradient we have chosen. Secondly red nanoselenium can be captured under a spectrophotometer in the range between 180-200nm, whereas cck8 is detected at 450nm. Then, fluorescent staining refers to fuels that emit fluorescent light, and after absorbing UV first or visible light, can transform short wavelengths of light into longer wavelengths of visible light waves in a sparkling vibrant colour. Therefore, when viewed through a fluorescence microscope, only the ROS labelled with fluorescence can be seen, and in our experiments, we really did not suffer from the interference of selenium nano-suspension in the colours.]

Comments 26:[The concentrations of SeNPs that were evaluated for their anti-antioxidant effect should be indicated in the main text of the manuscript (i.e. line 297) as well as in the Figure 2 caption.]

Response 26:[ Thank you for your advice. We have added specific concentration ranges to the manuscript]

Comments 26:[The reviewer also wants to know how the fluorescent signal related to the ROS quantification was processed by the authors and if this signal was normalized considering the number of cells. Additionally, it would be beneficial for the manuscript if the authors could include not only a representative picture of the in vitro ROS-related experiments but also a quantitative quantification of these experiments.]

Response 26:[ Thank you very much for suggesting it. We will decide on the period of time to test the cells based on the cell growth curve and also ensure that the concentration range is close by testing the od value. Finally, fluorescence staining for ROS will be performed. the data of ROS fluorescence quantification processed by image J will be uploaded in the Supplementary Material.]

Comments 27:[Information related to Figure 3, panels D, E, and F should be incorporated in the main text of the manuscript.]

Response 27:[ Thank you very much for your suggestions, these metrics have been added to the manuscript]

Comments 28:[The subheading title 4.3.1.2. should be rephrased into a more specific title that summarizes the findings indicated in the main text of the subheading.]

Response 28:[ Thank you very much for your suggestion, it has been changed to "Bone densitometry results in mice".]

Comments 29:[ The reviewer wants to know the biological relevance of evaluating the BMC and BMD upon exposure to the PS-MPs. Is there any indication that the PS-MPs can affect these read-outs?]

Response 29:[ Thank you very much for your question. We have not found any signs of this in our experiments. However, we suspected that MPs would affect the growth and development of the mice leading to their bone dysplasia, so we made an attempt to test it.]

Comments 30:[The authors should clearly indicate why the highest dose of PS-MPs was selected for subsequent in vivo experiments. The sentence indicated in lines 333 and 334 is vague in this regard.]

Response 30:[Thank you for your question. I am very sorry for not being able to explain this in the most appropriate section. After testing the liver biochemistry in mice, we found that high doses of MPs had the greatest impact on the index results. So we had decided then to follow up with high dose MPs to continue the experiment. We have removed the last sentence of the section.]

Comments 31:[In Figure 4, panel A, the reviewer cannot discern the condition depicted in the image. To provide clarity, a more detailed description should be included in the figure caption.]

Response 31:[ Thank you very much for your suggestion, it has been changed to "Radiographic images of mice under bone densitometry".]

Comments 32:[In subheading 4.3.2.1, the authors characterize their findings as 'significant' (i.e., line 345). However, they present only a representative liver histology image in Figure 5. Does the authors' analysis include quantitative data that supports their conclusions? If so, this quantitative data should be included in Figure 5.]

Response 32:[ Thank you very much for suggesting it. We are very sorry that our description was too deliberate and subjective, we have reassessed it in the light of the image and changed it to "relatively attenuated,"]

Comments 33:The reviewer would like to know how the authors quantified the fat tissue content depicted in Figure 6, panel H, as well as in line 379 of the main text.

Response 33:[ Thank you for your question. This is actually data from the bone density test, but for ease of discussion we have integrated it in this section. Bone densitometers measure the density of tissues in a living organism by means of x-rays, and the density of adipose tissue compared to the density of other tissues gives the data.]

Comments 34:[   The reviewer is curious as to why the authors referred to the NRLP3 inflammasome pathway as an inflammatory vesicle pathway.]

Response 34:[ We are very sorry for your query due to our misrepresentation. It has been changed to "NRLP3 inflammasome pathway".]

Comments 35:[The reviewer is uncertain about the number of animals included in the western blot quantification depicted in Figure 7, panels C-I.]

Response 35:[ We apologise for your query due to an oversight on our part. We are randomly selecting 3 liver samples from each group of 6 liver samples for wb experiments. Overall 3 biological replicates were performed]

Comments 36:[To enhance the manuscript, the authors should place the subheading 4.3.2.5 before 4.3.2.4. This arrangement would allow the authors to initially present the transcriptional effects of the treatments and subsequently describe their translational effects.]

Response 36:[ Thank you for your advice. We have switched the order of these two subsections]

 Comments 37:[The content in line 413 requires correction, as several of the targets mentioned by the authors are translational modifications that cannot be observed at the gene expression level (e.g., cleaved caspase 1 or phosphorylated NF-KB).]

Response 37:[ Thank you very much for the reminder. We have removed these two genes from the original article]

 Comments 38: [It would benefit the manuscript if the subheading 4.3.2.6 provided more information about the biological function of the selected mitochondrial genes in the main text.]

Response 38:[Thank you very much for your advice. A description of the mitochondrial body-related gene has been added to the text]

 Comments 39:[In the Discussion portion of the manuscript, several paragraphs require revision, shortening, and clarification to ensure that the focus is primarily on comparing the obtained results with previous reports, rather than providing general information.]

Response 39:[ Thank you for your suggestion. Emphasis has been placed on comparison with past reports.]

 Comments 39:[indings related to the BMC and BMD readouts as well as the total content of fact upon exposure to the PS-MPs are not discussed.]

Response 39:[ Thank you very much for your suggestion, as the BMD results do not correlate well with exposure to MPs vs. SeNPs. We then did not carry out much discussion.]

 Comments 39:[The reviewer has questioned why additional in vitro experiments were not conducted to evaluate the potential impact of the PS-MPs on the pro-inflammatory and mitochondrial pathways.]

Response 39:[ Your questions are very good. When we designed the experiment, we divided it into two main parts, one was to investigate the inflammatory damage induced by microplastics in the liver as well as the abnormalities in mitochondrial dynamics, and the second was to investigate the effects of microplastics on the lipid metabolism aspects of the liver, and the second part has already been published in a paper. The initial focus of the study was on in vivo experiments. In order to confirm the cytotoxicity of microplastics and the effect of selenium nanoparticles on cells, cellular experiments were carried out first. After determining the cytotoxicity of the microplastics, the in vivo experiments were carried out, so we did not do much experimental design and discussion on the molecular mechanism of the cells. This may be a good opportunity for us to find out more specific and detailed molecular mechanisms of microplastics on cells in our subsequent studies.

 Comments 40:[The manuscript presents several issues that require attention, including typographical errors (e.g. "caspase1" in line 400, etc.), inconsistencies (e.g. 4 weeks vs 28 days, numbers written in numerical characters vs full letters, etc.), incorrect abbreviations, scientific notation (e.g. centrifuge speed, cell densities, etc.), and non-scientific language (e.g. "revived cells," "animals were contaminated," etc.). It is recommended that the authors carefully review and revise the manuscript to address these concerns]

Response 40:[ All "casepase1" have been changed to "caspase1". It has been made consistent. All centrifuge speed units have been changed to rpm. The units for cell density have been changed to normal. The words "revived cells" have been deleted. The words  "animals were contaminated,"have been deleted]

 Comments 41:[The names of the genes should be revised and written according to the scientific guidelines.]

Response 41:[ Thank you for your advice..All "casepase1" have been changed to "caspase1".]

 Comments 42: [The font and size of the Figure 5 caption should be modified and uniformized with the rest of the figure’s captions of the manuscript.]

Response 42:[ Thank you for your suggestion, it has been standardised in font size and typography]

 Comments 43: [The figure captions should also be revised to include the sample size, including biological and technical replicates, the statistical test used for comparisons, and references to the scale bar in all captions.]

Response 43:[ Thanks to your suggestion, the sample size has been added to the description of all charts]

  Comments 44: [The tense of the Materials and Methods section should be changed to past tense]

Response 44:[ Thank you for your advice. The tense of the Materials and Methods section had been changed to past tense]

[1]Cunningham EM,Sigwart JD. Environmentally Accurate Microplastic Levels and Their Absence from Exposure Studies. Integr Comp Biol. 2019;59 (6):1485-1496.

[2]Yin K,Wang D,Zhang Y, et al. Polystyrene microplastics promote liver inflammation by inducing the formation of macrophages extracellular traps. J Hazard Mater. 2023;452:131236.

[3]Shi C,Han X,Guo W, et al. Disturbed Gut-Liver axis indicating oral exposure to polystyrene microplastic potentially increases the risk of insulin resistance. Environ Int. 2022;164:107273.

[4]K C, P.B,Maharjan, A.,Acharya, M.,Lee, D.,Kusma, S.,Gautam, R.,Kwon, J.T,Kim, C.,Kim, K.,Kim, H., & Heo, Y. (2023). Polytetrafluorethylene microplastic particles mediated oxidative stress, inflammation, and intracellular signaling pathway alteration in human derived cell lines. The Science of the total environment, 897, 165295.

[5]Danopoulos E, Twiddy M, West R, Rotchell JM. A rapid review and meta-regression analyses of the toxicological impacts of microplastic exposure in human cells. J Hazard Mater. 2022 Apr 5;427:127861.

[6]Li H, Yang X, Zhang J, Liu J, Fu Y, Kyin S, Zhang M, Peng Y, Zhou D. Selenium nanoparticles reduce cardiomyocyte apoptosis in Ascites Syndrome in Broiler Chickens via the ATF6-DR5 signaling pathway

[7]Ye, X.Q,Zhu, Y.R,Yang, Y.Y,Qiu, S.J, & Liu, W.C (2023). Biogenic Selenium Nanoparticles Synthesized with Alginate Oligosaccharides Alleviate Heat Stress-Induced Oxidative Damage to Organs in Broilers through Activating Nrf2-Mediated Anti-Oxidation and Anti-Ferroptosis Pathways. Antioxidants (Basel, Switzerland), 12 (11), .

[8]Abdel-Gaber, R.,Hawsah, M.A,Al-Otaibi, T.,Alojayri, G.,Al-Shaebi, E.M,Mohammed, O.B,Elkhadragy, M.F,Al-Quraishy, S., & Dkhil, M.A (2023). Biosynthesized selenium nanoparticles to rescue coccidiosis-mediated oxidative stress, apoptosis and inflammation in the jejunum of mice. Frontiers in immunology, 14, 1139899.

Reviewer 4 Report

Comments and Suggestions for Authors

Comments to the Authors of manuscript number nutrients-3129832 entitled “Nano-selenium modulates NF-κB/NLRP3 pathway and mitochondrial dynamics to attenuate microplastic-induced liver injury

The study found that nano-selenium particles (SeNPs) can alleviate liver damage caused by microplastics (PS-MPs). In vitro experiments showed that SeNPs promote the growth of healthy mouse liver cells, whereas microplastics have a harmful effect, reducing cell count. In vivo experiments on mice confirmed that SeNPs reduce the detrimental effects of PS-MPs on the liver by inhibiting the NF-κB/NLRP3 inflammatory signaling pathway. SeNPs improve mitochondrial function, which helps prevent metabolic disorders and inflammation. The findings suggest that SeNPs could be useful in mitigating inflammatory injuries caused by microplastics.

1. L 49- The phrase "natural protective factor against liver disease" needs more precise scientific validation

2. L 51- Jöns Jacob Berzelius?

3. L 52- The statement "selenium plays a protective role in organisms in the form of selenoprotein"is unclear.

4. L 55 - the liver serves as the storage site for selenium? Really?

5. L 56- “the liver has a considerably elevated selenium concentration compared to other tissues and organs” needs specific references or data for support

6. L 59- uncler, because selenium nanoparticles can vary in form and color depending on synthesis and context.

7. the lack of hypothesis

8. Line 70: The exact starting materials and quantities for the synthesis of nano-selenium are not specified.

9. Line 71: The conditions for agitation (e.g., temperature, shaking speed) are not detailed

10. Lines 78-79: The repetition of "at 4 degrees Celsius" is redundant and confusing. Additionally, the precise concentration of Tris-HCl used is not mentioned.

11. Line 83: The phrase "Mix thoroughly" lacks specifics

12. Line 87: The phrase "When the time comes" is vague and could be replaced with a more precise timeline or condition

13. Lines 87-88: The freeze-drying process needs more detail on the specific parameters used (e.g., temperature, pressure).

14. Lines 107-108: The exact procedure for treating cells with different concentrations of PS-MPs and SeNPs is not described (e.g., how the treatments are prepared and applied).

15. L 111-112-the concentration and duration of trypsinization are needed

16. Lines 115-116: Clarification is needed on how "independent samples were taken in triplicate" – does this mean three independent experiments or three technical replicates within one experiment?

17. Lines 120-121: More information is needed on how "the area occupied by the cells reaches 80-90% of the bottom area of the T25 bottle" is determined

18. Line 123: The preparation of PS-MPs at different concentrations should be explained in detail, including any solvents or vehicles used.

19. Line 126: The measurement of absorbance needs more specifics, including the wavelength used for detection and the equipment settings.

20. Line 127: The phrase "three replicates using biological samples" should specify whether these are technical or biological replicates and how consistency between replicates is ensured.

21. Lines 131-132: Similar to the previous section, how "the area occupied by the cells reaches 80-90% of the bottom area of the T25 bottle" is determined should be clarified.

22. Line 133: The term "inductively coupled method" for introducing SeNPs needs explanation

23. Line 135: The concentration of the CCK8 solution is needed

24. Lines 136-137: The measurement of absorbance should include details on the equipment used and any calibration or specific settings.

25. Line 137: The phrase "three replicates using biological samples" should specify whether these are technical or biological replicates

26. L 142- how PS-MPs and SeNPs are prepared and added

27. L 149- the total number of animals is needed

28. L 154- the study design is unclear and how many mice was per each group

29. L 158- what type of diet were fed?

30. Line 161, 171: The term "contaminated by gavage" could be clearer if rephrased to "administered by gavage."

31- What was given to the control group?

32. Line 162: The specific criteria for observing "fluctuations in their weight" and "consumption of food and water" should be clarified (e.g., how these are measured and recorded).

33. Line 163: The concentration and volume of sodium pentobarbital used for euthanasia should be specified.

34. Lines 166-168: how the liver tissue is "pulverized into a uniform mixture," prepared for histopathological sections, and the exact conditions for storage at -80°C.

35. Line 179: The concentration and volume of sodium pentobarbital used for euthanasia need to be specified again

36. part 2.5.2.1. Histopathological analysis- what parameters were assessed

37. 2.5.2.2. Transmission electron Microscopy - what parameters were assessed

38. L 203- is “The BS-240VET (Myriad 203 Biologicals Ltd., Shenzhen, China)” analyser?

39. what production were the reagents?

40. L 207- the study design is so unclear, it was not know what tissues were collected

41. L 211- where?

42. L 217- what primary antibody? Secondary antibody?

43. part 2.5.2.7. – how many technical repetitions/

44. L 238- primary antibodies are not given

45. L 241- secondary antibodies?

46. L 257- how the average particle size was calculated (e.g., number of particles measured, software used).

47. L 258-259- how many particles were measured to obtain this average size and the method used for measurement.

48. Figure 1, E,F, G,H -unclear, too small

49. L 272- it was not described in material and methods

50. L 288- More details on the experimental setup are needed, such as the exact number of replicates and the control conditions used.

51. Line 289: The term "various concentrations" should specify the exact concentrations tested to clarify the range of the experiment.

52. Line 303: how the fluorescence intensity was quantified and how it was determined that the intensity in the PS-MPs group exceeded that of the Control group.

53. Line 305: The mention of "weaker" fluorescence intensity in the Control and SeNPs groups should be quantified and compared with statistical results.

54. Lines 310-311: Define how the liver was identified as a target organ for PS-MPs

55. Line 313: "10 mg/mL of PS-MPs" is a concentration or a dose

56. Line 338: The phrase "Initially, we evaluated the inhibitory impact of SeNPs" should specify the exact method or parameters used to evaluate the inhibitory impact

57. L 339- what criteria were used to define "typical" morphology

58. Lines 342-343: how disorganized hepatocyte arrangement, cytoplasmic vacuolization, and inflammatory cell infiltration were quantified or assessed. more information on how the images were analyzed are needed.

59. L 346-347- what constitutes "greatly reduced" cytoplasmic vacuolation and how it was measured or quantified

60. Figure 5 , part A is poorly described, illegible

61. Figure 7 , part A illegible

62. Figure 7, part B, the bound of caspase 1 illegible, it should be repeated

63. L 429-431- how microplastics specifically impair the liver?

64. L 433-434- The phrase “rare study” is vague

65. Lines 454-459-  It is not clear how SeNPs specifically interact with GPx and the broader antioxidant system in the liver.

66. Lines 494-507 How do these pathways interact with oxidative stress induced by PS-MPs?

Author Response

 Thank you very much for your valuable comments on this manuscript, I will highlight the changes in purple!

Comments 1:[L 49- The phrase "natural protective factor against liver disease" needs more precise scientific validation]

Response 1:[Selenium plays an important role in anti-oxidative stress and anti-inflammatory regulation and is regarded as a crucial micronutrient that is indispensable for human well-being]

Comments 2:[L 51- Jöns Jacob Berzelius?]

Response2:[Yes!He discovered and made for the first time several elements such as silicon, thorium and selenium]

Comments 3:[The statement "selenium plays a protective role in organisms in the form of selenoprotein"is unclear]

Response3:[Thank you very much for your suggestion, we have changed this formulation to make it clearer.The necessity of Se for living organisms is due to the fact that it is a component of selenocysteine (Sec) found in various selenoproteins. Selenoproteins are synthesized in the cells via a unique mechanism that involves specific enzymes and factors and directly depends on Se intake. Human selenoproteome is encoded by 25 selenoprotein genes. The functions of the encoded proteins in the human body are extremely diverse. Many selenoproteins have a pronounced]

Comments 4:[L 55 - the liver serves as the storage site for selenium? Really?]

Response 4:[Thank you very much for pointing out our mistake, we wanted to express that selenium is mostly metabolised to selenoproteins in the liver, so the liver is the main site of selenium metabolism. We have changed the inaccurate part of the original text.]

Comments 5:[L 56- “the liver has a considerably elevated selenium concentration compared to other tissues and organs” needs specific references or data for support]

Response 5:[Please forgive our uncritical presentation, after a full reading of the references. We have found that the liver will take up and absorb selenium well relative to other tissues. Changes have been made to the original text.]

Comments 6:[L 59- uncler, because selenium nanoparticles can vary in form and color depending on synthesis and context.]

Response 6:[You are right, we will change what is not rigorous. "SeNPs are tiny particles of red selenium" to "SeNPs are tiny particles of selenium"]

Comments 7: [the lack of hypothesis]

Response 7:[I'm sorry we didn't understand what you meant.]

Comments 8:[Line 70: The exact starting materials and quantities for the synthesis of nano-selenium are not specified.]

Response 8:[Thanks for your input, we used a 20 mmol/L sodium selenite solution and 1% Bacillus licheniformis with OD600=0.5. The above has been added to the manuscript.]

Comments 9:[Line 71: The conditions for agitation (e.g., temperature, shaking speed) are not detailed]

Response 9:[Thank you for your comments. The shaking conditions were 180 r/min, 37°C. We have added this to the manuscript]

Comments 10:[Lines 78-79: The repetition of "at 4 degrees Celsius" is redundant and confusing. Additionally, the precise concentration of Tris-HCl used is not mentioned]

Response 10:[Thanks for the heads up, we've removed the duplicates from the content and added the specific concentrations]

Comments 11:[ Line 83: The phrase "Mix thoroughly" lacks specifics]

Response 11:[ Thank you for pointing out the unclear part of the manuscript, which has been corrected to read "well-mixed".]

Comments 12:[ Line 87: The phrase "When the time comes" is vague and could be replaced with a more precise timeline or condition]

Response 12:[ Thank you for your suggestion, the time representation in the methodology has been changed to be more specific]

Comments 13:[Lines 87-88: The freeze-drying process needs more detail on the specific parameters used (e.g., temperature, pressure).]

Response 13:[ Thanks to your suggestion, specific parameters for freeze-drying have been added to the manuscript]

Comments 14:[ Lines 107-108: The exact procedure for treating cells with different concentrations of PS-MPs and SeNPs is not described (e.g., how the treatments are prepared and applied).]

Response 14:[ Thanks to your suggestion, the method of preparing MPs solutions for SeNPs solutions has been added into 2.4]

Comments 15:[ L 111-112-the concentration and duration of trypsinization are needed]

Response 15:[ Thank you for your advice, we use the White Shark brand of 0.25% trypsin and follow its instructions for experimentation. Have added the above to the manuscript]

Comments 16:[Lines 115-116: Clarification is needed on how "independent samples were taken in triplicate" – does this mean three independent experiments or three technical replicates within one experiment?]

Response 16:[ Your query is not unreasonable. We have modified this to three biological repetitions]

Comments 17:[ Lines 120-121: More information is needed on how "the area occupied by the cells reaches 80-90% of the bottom area of the T25 bottle" is determined]

Response 17:[ Thank you for asking questions about this. We use an inverted microscope to look at the bottom of the bottle to see if the cells occupy 80-90 per cent of the bottom area of the bottle.]

Comments 18:[Line 123: The preparation of PS-MPs at different concentrations should be explained in detail, including any solvents or vehicles used]

Response 18:[ Thank you for your advice. We have added the preparation of different concentrations of MPs solutions to the manuscript]

Comments 19:[Line 126: The measurement of absorbance needs more specifics, including the wavelength used for detection and the equipment settings.]

Response 19:[ Thank you for your advice. We are using a Spectro star Nano enzyme marker at 450nm to observe]

Comments 20:[Line 127: The phrase "three replicates using biological samples" should specify whether these are technical or biological replicates and how consistency between replicates is ensured.]

Response 20:[ Thank you for your advice."three replicates using biological samples" means“biological replicates”.]

Comments 21:[Lines 131-132: Similar to the previous section, how "the area occupied by the cells reaches 80-90% of the bottom area of the T25 bottle" is determined should be clarified.]

Response 21:[ Your suggestions will help me revise the manuscript better. The answer to this question is the same as Comments 17]

Comments 22:[Line 133: The term "inductively coupled method" for introducing SeNPs needs explanation]

Response 22:[ We apologise for the confusion caused by our misrepresentation. We have amended our original comment and deleted "inductively coupled method".]

Comments 23: [Line 135: The concentration of the CCK8 solution is needed]

Response 23:[ Thank you for your advice. We used the standard cck8 reagent supplied by white shark and did not change its concentration additionally]

Comments 24:[Lines 136-137: The measurement of absorbance should include details on the equipment used and any calibration or specific settings]

Response 24: [Thanks for the heads up. We have added the detailed model number of the enzyme labeller (Spectro star Nano) to the manuscript]

Comments 25:[Line 137: The phrase "three replicates using biological samples" should specify whether these are technical or biological replicates]

Response 255: [Thank you for your advice."three replicates using biological samples" means“biological replicates”. This has been modified.]

Comments 26:[L 142- how PS-MPs and SeNPs are prepared and added]

Response 26:[ Thank you for your advice. Preparation and addition of microplastics and nanoselenium have been added to the manuscript]

Comments 27:[ L 149- the total number of animals is needed]

Response 27:[ Thank you for your suggestion, the number of mice has been added to the manuscript.]

Comments 28:[L 154- the study design is unclear and how many mice was per each group]

Response 28:[ Thank you for your advice, We divided the mice into four groups of six mice each. The above was added to the manuscript]

Comments 29:[L 158- what type of diet were fed?]

Response 29:[ Thank you for your question. The experimental grade rat food we normally provide for mice]

Comments 30:[Line 161, 171: The term "contaminated by gavage" could be clearer if rephrased to "administered by gavage."]

Response 30:[ Thank you very much for your advice. We have changed "contaminated by gavage" to "administered by gavage."]

Comments 31:[What was given to the control group?]

Response 31:[ It's an honour to answer your questions. Unfortunately we did not do anything with the blank group.]

Comments 32:[Line 162: The specific criteria for observing "fluctuations in their weight" and "consumption of food and water" should be clarified (e.g., how these are measured and recorded).]

Response 32:[ Thank you very much for your advice. We will weigh and record the mice weekly. However, observations of food and water consumption will be limited to weighing them after confirming that they have consumed all their food and water. Therefore, we believe that this description is not rigorous enough, and we have deleted the "food and water consumption" section.]

Comments 33:[Line 163: The concentration and volume of sodium pentobarbital used for euthanasia should be specified.]

Response 33:[ Thank you for your question, we use 5% sodium pentobarbital at a dose of 200mg/kg to euthanise mice.]

Comments 34:[ Lines 166-168: how the liver tissue is "pulverized into a uniform mixture," prepared for histopathological sections, and the exact conditions for storage at -80°C]

Response 34:[ Thank you for your advice. This has been removed, and each experimental liver treatment is described accordingly in its corresponding subsection]

Comments 35:[Line 179: The concentration and volume of sodium pentobarbital used for euthanasia need to be specified again]

Response 35:[ Thank you for your advice. Concentration of sodium pentobarbital has been added to the manuscript]

Comments 36:[part 2.5.2.1. Histopathological analysis- what parameters were assessed]

Response 36:[ Thank you for your question. The results and content of the pathological assessment of liver sections are described in detail in 4.3.3. In fact, as a result of your question, we have thought that the title should be simpler and clearer and have changed the title of this section to "H.E. staining of liver".]

Comments 37:[2.5.2.2. Transmission electron Microscopy - what parameters were assessed]

Response 37:[ Thank you for your question. Similar to the reply to the previous comment, TEM's observations on liver sections are in subsection 4.3.3. Meanwhile, I have changed the title of this section to "2.5.4 Blood biochemical analysis".]

Comments 38:[L 203- is “The BS-240VET (Myriad 203 Biologicals Ltd., Shenzhen, China)” analyser?]

Response 38:[yes!We have changed "myraid" to "mindray" in the text.]

Comments 39:[what production were the reagents?]

Response 39:[ We're using the same reagents from mingray.]

Comments 40:[L 207- the study design is so unclear, it was not know what tissues were collected]

Response 40:[ Thank you for your question. We only took liver tissue for this experiment, the measurement of bone density can be done on a bone densitometer without expelling the bone tissue, which is based on the principle that x-rays detect the density of bone and soft tissue and calculate the ratio.]

Comments 41:[L 211- where?]

Response 41:[ I'm sorry it may be due to some bug that prevents you from seeing L211]

Comments 42:[what primary antibody? Secondary antibody?]

Response 42:[ Thank you for your question. The primary antibody were NLRP3 Rabbit pAb (A5652) and NFKB1 Rabbit pAb (A6667) both from ABclonal . The secondary antibodies we used was the iNOS Rabbit mAb (A3774) from ABclonal]

Comments 43:[ part 2.5.2.7. – how many technical repetitions/]

Response 43:[ Thank you for your question. There are three technical repeats for each RNA test]

Comments 44:[ L 238- primary antibodies are not given]

Response 44:[ Thank you for your question, the primary and secondary antibodies are explained at the end of this paragraph]

Comments 45:[ secondary antibodies?]

Response 45:[ The secondary antibody was HRP-conjugated Goat anti-Rabbit IgG (H+L) (AS014) purchased from ABclonal. The above has been added to the manuscript]

Comments 46: [how the average particle size was calculated (e.g., number of particles measured, software used)

Response 46:[ Thank you for your question. Our school's electron microscope comes with a scale. We are randomly selecting all the selenium nanoparticles in 3 fields of view and then measuring their approximate particle size via the scale.]

Comments 47:[L 258-259- how many particles were measured to obtain this average size and the method used for measurement.]
Response 47:[ We randomly examined selenium nanoparticles in three fields of view under the electron microscope, and the particle size of selenium nanoparticles was calculated by means of a scale.]

Comments 48:[Figure 1, E,F, G,H -unclear, too smal]

Response 48:[ We apologise for any inconvenience caused by typographical problems, but we will upload every high-resolution image to the journal for readers' convenience.]

Comments 49:[L 272- it was not described in material and methods]

Response 49:[Thank you for your review. L272 in the manuscript corresponds to 2.4.1.]

Comments 50:[L 288- More details on the experimental setup are needed, such as the exact number of replicates and the control conditions used.]

Response 50:[ Thank you for your question. Details on this section have been revised and added in Materials and Methods]

Comments 51:[ Line 289: The term "various concentrations" should specify the exact concentrations tested to clarify the range of the experiment.]

Response 51:[ Thank you for your advice. Specific different concentrations have been added to the manuscript.]

Comments 52:[Line 303: how the fluorescence intensity was quantified and how it was determined that the intensity in the PS-MPs group exceeded that of the Control group.]

Response 52:[ Thank you very much for your question. We are using IMAGE j software to quantify the fluorescence intensity in the images. Specific fluorescence data will be uploaded to the supplementary material.]

Comments 53:[  Line 305: The mention of "weaker" fluorescence intensity in the Control and SeNPs groups should be quantified and compared with statistical results.]

Response 53:[ Thank you for your question. Actually the naked eye can see that the fluorescence intensity and staining area of the SeNPs group and the control group are weaker relative to the other groups. We also analysed them with IMAGE J software and the results have been uploaded in the Supplementary file。]

Comments 54:[Lines 310-311: Define how the liver was identified as a target organ for PS-MPs]

Response 54:[Thank you for your question. The description is not rigorous enough, and although MPs can accumulate in the liver via the hepatic-intestinal axis and cause damage, there is no evidence to suggest that the liver is a target organ for MPs. We have removed this sentence.]

Comments 55:[Line 313: "10 mg/mL of PS-MPs" is a concentration or a dose]

Response 55:[ Thank you for your question. "10mg/ml" means 10mg/ml of microplastic solution.]

Comments 56:[Line 338: The phrase "Initially, we evaluated the inhibitory impact of SeNPs" should specify the exact method or parameters used to evaluate the inhibitory impact]

Response 56:[ Thank you for your question. There shouldn't be overly subjective judgements about you in the results section. We have removed this sentence.]

Comments 57:[L 339- what criteria were used to define "typical" morphology]

Response 57:[I apologise for your query due to a writing error. We have changed "typical liver morphology" to "normal hepatocyte morphology".]

Comments 58:[Lines 342-343: how disorganized hepatocyte arrangement, cytoplasmic vacuolization, and inflammatory cell infiltration were quantified or assessed. more information on how the images were analyzed are needed]

Response 58: [Thank you for your question. We observed normal hepatocyte sections under an electron microscope for comparison. Literature was also reviewed to confirm the morphology of normal liver sections. We are not aware of any specific quantitative criteria for liver sections, so please forgive us.]

Comments 59:[ L 346-347- what constitutes "greatly reduced" cytoplasmic vacuolation and how it was measured or quantified]

Response 59:[ Thank you for your suggestion. After careful consideration of our results, we believe that the phrase "greatly reduced" is not rigorous enough, so we have changed it to "relative reduction in cytoplasmic vacuolisation". We hypothesise that the morphological changes in the hepatocytes are due to inflammatory damage caused by the accumulation of hepatic energy.]

Comments 60:[figure 5 , part A is poorly described, illegible]

Response 60:[ Thank you for your review. It has been changed to H.E stained images and electron microscopic images of liver tissue sections.]

Comments 61:[Figure 7 , part A illegible]

Response 61:[ Thank you for your advice. We'll upload the original high-resolution images for easy access!]

Comments 62:Figure 7, part B, the bound of caspase 1 illegible, it should be repeated

Response 62:[ Thank you very much for your advice. Possibly due to the quality of the antibodies, we found that the trends were recognisable but the images were slightly blurred, and we apologise for any inconvenience caused to your review.]

Comments 63:[L 429-431- how microplastics specifically impair the liver?]

Response 63:[ Thank you for the question.Accumulation of microplastics causes oxidative stress, inflammation and apoptosis-related toxic effects in the liver, as well as disturbances in glucose and lipid metabolism in the liver.]

Comments 64:[L 433-434- The phrase “rare study” is vague]

Response 64:[ Thank you for your advice. We have amended this]

Comments 65:Lines 454-459-  It is not clear how SeNPs specifically interact with GPx and the broader antioxidant system in the liver

Response 65:[ Thank you for your question. Selenium performs its biological function in the body in the form of selenoproteins.Selenium is mainly found in the body as selenocysteine (selenocysteine, Sec) and selenomethionine (Se-Met).Selenium binds to proteins in the body mainly in two forms: selenocysteine (Sec) and selenomethionine (Se-Met). Sec is a translational act mediated by the codon UGA, while Se-Met is involved in protein molecules by random substitution of methionine, most likely due to the tolerance of methionyl-tRNA synthetase to its incorporation into proteins. And selenium as a raw material for the synthesis of five glutathione peroxidases plays an important role in the antioxidant system of the organism.

Comments 66:[Lines 494-507 How do these pathways interact with oxidative stress induced by PS-MPs?]

Response 66:[ Thank you for your question. Oxidative stress refers to a state of imbalance between oxidative and antioxidant effects in the body. Excessive ROS production by hepatocytes due to MPs is a manifestation of oxidative stress. However, the main focus of the discussion in this manuscript is not oxidative stress, but the mechanism of hepatic inflammation and mitochondrial damage caused by MPs.]

 Thank you very much for your valuable comments on this manuscript, I will highlight the changes in purple!

Comments 1:[L 49- The phrase "natural protective factor against liver disease" needs more precise scientific validation]

Response 1:[Selenium plays an important role in anti-oxidative stress and anti-inflammatory regulation and is regarded as a crucial micronutrient that is indispensable for human well-being]

Comments 2:[L 51- Jöns Jacob Berzelius?]

Response2:[Yes!He discovered and made for the first time several elements such as silicon, thorium and selenium]

Comments 3:[The statement "selenium plays a protective role in organisms in the form of selenoprotein"is unclear]

Response3:[Thank you very much for your suggestion, we have changed this formulation to make it clearer.The necessity of Se for living organisms is due to the fact that it is a component of selenocysteine (Sec) found in various selenoproteins. Selenoproteins are synthesized in the cells via a unique mechanism that involves specific enzymes and factors and directly depends on Se intake. Human selenoproteome is encoded by 25 selenoprotein genes. The functions of the encoded proteins in the human body are extremely diverse. Many selenoproteins have a pronounced]

Comments 4:[L 55 - the liver serves as the storage site for selenium? Really?]

Response 4:[Thank you very much for pointing out our mistake, we wanted to express that selenium is mostly metabolised to selenoproteins in the liver, so the liver is the main site of selenium metabolism. We have changed the inaccurate part of the original text.]

Comments 5:[L 56- “the liver has a considerably elevated selenium concentration compared to other tissues and organs” needs specific references or data for support]

Response 5:[Please forgive our uncritical presentation, after a full reading of the references. We have found that the liver will take up and absorb selenium well relative to other tissues. Changes have been made to the original text.]

Comments 6:[L 59- uncler, because selenium nanoparticles can vary in form and color depending on synthesis and context.]

Response 6:[You are right, we will change what is not rigorous. "SeNPs are tiny particles of red selenium" to "SeNPs are tiny particles of selenium"]

Comments 7: [the lack of hypothesis]

Response 7:[I'm sorry we didn't understand what you meant.]

Comments 8:[Line 70: The exact starting materials and quantities for the synthesis of nano-selenium are not specified.]

Response 8:[Thanks for your input, we used a 20 mmol/L sodium selenite solution and 1% Bacillus licheniformis with OD600=0.5. The above has been added to the manuscript.]

Comments 9:[Line 71: The conditions for agitation (e.g., temperature, shaking speed) are not detailed]

Response 9:[Thank you for your comments. The shaking conditions were 180 r/min, 37°C. We have added this to the manuscript]

Comments 10:[Lines 78-79: The repetition of "at 4 degrees Celsius" is redundant and confusing. Additionally, the precise concentration of Tris-HCl used is not mentioned]

Response 10:[Thanks for the heads up, we've removed the duplicates from the content and added the specific concentrations]

Round 2

Reviewer 2 Report

Comments and Suggestions for Authors

The authors replied to all the questions of this reviewer.

Author Response

The authors replied to all the questions of this reviewer.

Reviewer 3 Report

Comments and Suggestions for Authors

Dear Editor and Authors,

I would like to thank the authors for their efforts in revising the manuscript and addressing the reviewer's comments. While the recent improvements to the manuscript are commendable, several issues still need to be addressed before it can be published in Nutrients.

·       In the abstract, the authors claim that microplastics have a detrimental impact on normal mouse hepatocyte cell suspensions, resulting in a decrease in cell count. Do the authors evaluate the viability of the PS-MPs in the cellular suspension or after the cells have already adhered to the cell culture plate? If cellular viability was assessed after the cells adhered to the plate, this assertion in the abstract should be revised.

·       Please remove "EE et al 2021" from line 46 of the main manuscript.

·       To enhance the reproducibility of the protocols described in the materials and methods section, RPM should be indicated as x g.

·       The subheading 2.4.1. Culture of mouse hepatocytes should be rephrased (the authors currently describe this subheading as a laboratory protocol). Furthermore, the last sentence of that subheading (i.e., line 134) should be removed as it is redundant.

·       The information indicated in lines 146-149 can be excluded from the manuscript.

·       In line 155, what do the authors mean by 5 micrometers? Do they intend to indicate 5 microliters?

·       The secondary antibody indicated by the authors in line 243 (i.e., iNOS Rabbit mAb (A3774)) is a primary antibody. The actual secondary antibody used by the authors should be included in the manuscript.

·       The title of the subheading 2.5.8 'Quantitative real-time quantitative fluorescence PCR' should be rephrased to the actual name of the technique (e.g., 'real-time quantitative PCR').

·       The information provided in lines 252-255 is not necessary for the manuscript and should be removed or rephrased.

·       Despite the authors' efforts to address the reviewer's concerns regarding the sample size utilized in the in vivo experiments, it is still necessary to include the sample size in the figure captions for Figures 3, 6, 7, and 8. Additionally, the caption for Figure 5 should indicate that the images presented are representative of each group.

·       While the authors have provided information about the function of MFN1 and MFN2 genes, there is no mention of the other genes involved in mitochondrial dysfunction. Therefore, the authors should either provide additional information about these genes in the main manuscript or remove the information about MFN1 and MFN2 genes.

·       The term "provided guidelines" in line 268 is unclear to the reviewer and should be rephrased and clarified in the main manuscript.

·       Subheadings 4.2.2 and 4.2.3 should be rephrased according to the new subheadings indicated in the materials and methods section (i.e., 2.4.3 and 2.4.4).

·       In Figure 4, panel A, the authors should indicate which group of animals is depicted in the images (e.g., control group).

Comments on the Quality of English Language

·       The tense of the materials and methods section is not uniform. Some sentences are written in the past tense, while others are written in the present tense. This inconsistency should be addressed, and the tense should be standardized.

·       The manuscript contains several discrepancies concerning the designation of various parameters, such as the concentration of cells (see, for instance, the differences in how the authors referred to this parameter on lines 152 and 163). These inconsistencies should be rectified and standardized.

·       It is also recommended that the abbreviations be revised and standardized, indicating the full term only the first time it appears in the manuscript and subsequently using the acronym.

·       The names of the genes should be updated to conform to the international nomenclature guidelines.

Author Response

Comments 1:[  In the abstract, the authors claim that microplastics have a detrimental impact on normal mouse hepatocyte cell suspensions, resulting in a decrease in cell count. Do the authors evaluate the viability of the PS-MPs in the cellular suspension or after the cells have already adhered to the cell culture plate? If cellular viability was assessed after the cells adhered to the plate, this assertion in the abstract should be revised.]

Response1:[ Thank you very much for your reminding. We did examine the effect of MPs on cell growth, not on liver cell suspension. We modified it in the summary.]

Comments 2:[ Please remove "EE et al 2021" from line 46 of the main manuscript.]

Response2:[ Thank you very much for your careful review. We have removed "EE et al 2021" from the manuscript.]

Comments 3:[ To enhance the reproducibility of the protocols described in the materials and methods section, RPM should be indicated as x g.]

Response3:[ We very much agree with your proposed revision of the manuscript from the perspective of experimental reproduction, which I believe will greatly help improve the quality of our manuscript. However, the production process of SeNPs is determined according to the research results of our laboratory equipment and previous papers. For the part involving centrifugal speed, we directly used the production method of our laboratory predecessors, and in order to unify the consistency of the full text units, we did not replace the RPM with x g. We hope you can understand and thank you again for your help in improving the quality of our papers]

Comments 4:[The subheading 2.4.1. Culture of mouse hepatocytes should be rephrased (the authors currently describe this subheading as a laboratory protocol). Furthermore, the last sentence of that subheading (i.e., line 134) should be removed as it is redundant.]

Response4:[ We find your suggestion very worthy of reference. We have rewritten the title of 2.4.1 and removed the last sentence of that section]

Comments 5:[ The information indicated in lines 146-149 can be excluded from the manuscript.]

Response5:[ Thank you very much for your advice. We have removed the last sentence of 2.4.2 from the manuscript]

Comments 6:[ In line 155, what do the authors mean by 5 micrometers? Do they intend to indicate 5 microliters?]

Response 6:[ Thank you very much for pointing out our abbreviation error. "5 micrometers" means "5 um". We have amended the original text]

Comments 7:[The secondary antibody indicated by the authors in line 243 (i.e., iNOS Rabbit mAb (A3774)) is a primary antibody. The actual secondary antibody used by the authors should be included in the manuscript.]

Response 7:[ Thank you very much for your careful review. We have changed the primary antibody to the secondary antibody]

Comments 8:[ The title of the subheading 2.5.8 'Quantitative real-time quantitative fluorescence PCR' should be rephrased to the actual name of the technique (e.g., 'real-time quantitative PCR').]

Response 8:[ Thank you very much for your comments on the inappropriateness of our manuscript. We have changed the title of 2.5.8]

Comments 9:[ The information provided in lines 252-255 is not necessary for the manuscript and should be removed or rephrased.]

Response 9:[ Thank you very much for your advice. We have deleted the redundant parts from the manuscript]

Comments 10:[  Despite the authors' efforts to address the reviewer's concerns regarding the sample size utilized in the in vivo experiments, it is still necessary to include the sample size in the figure captions for Figures 3, 6, 7, and 8. Additionally, the caption for Figure 5 should indicate that the images presented are representative of each group.]

Response 10:[ Thank you very much for your thoughts on our manuscript. We are equally positive about the description of increasing the sample size in the picture. However, in consideration of the uniformity and aesthetics of the picture, we still give up this modification, for which we are deeply sorry. But at the same time, we have added a description of the sample size in the subtitle of the image so that readers can find it. We have indicated in the header of Figure 5 that the above images are representative of each group.]

Comments 11:[While the authors have provided information about the function of MFN1 and MFN2 genes, there is no mention of the other genes involved in mitochondrial dysfunction. Therefore, the authors should either provide additional information about these genes in the main manuscript or remove the information about MFN1 and MFN2 genes.]

Response 11:[ Thank you very much for your suggestions, but the description of the results in section 4.3.8 is really not complete. We have added the corresponding descriptions of each gene diagram to the manuscript.]

Comments 12:[  The term "provided guidelines" in line 268 is unclear to the reviewer and should be rephrased and clarified in the main manuscript.]

Response 12:[Thank you for your advice. We have changed "provided guidelines" to "Prospectus of WanLei Bio"]

Comments 13:[Subheadings 4.2.2 and 4.2.3 should be rephrased according to the new subheadings indicated in the materials and methods section (i.e., 2.4.3 and 2.4.4).]

Response 13:[ Thank you very much for your advice. We have changed title 4.2.2 to "Inhibitory effect of PS-MPs on hepatocyte proliferation" and at the same time we have changed title 4.2.3 to "Effect of SeNPs on hepatocyte proliferation" proliferation”]

Comments 14:[  In Figure 4, panel A, the authors should indicate which group of animals is depicted in the images (e.g., control group).]

Response 14:[ Thank you very much for your help in optimizing the details of our manuscript. We have changed the title of Figure 4 to "Control mice".]

Comments 15:[The tense of the materials and methods section is not uniform. Some sentences are written in the past tense, while others are written in the present tense. This inconsistency should be addressed, and the tense should be standardized.]

Response 15:[ Thank you very much for pointing out our mistakes in the use of tenses. We will check carefully and unify the tenses to improve the quality of our manuscripts]

Comments 16:[The manuscript contains several discrepancies concerning the designation of various parameters, such as the concentration of cells (see, for instance, the differences in how the authors referred to this parameter on lines 152 and 163). These inconsistencies should be rectified and standardized.]

Response 16:[ Thank you very much for your advice. We must check and unify the units carefully. We have changed "96-well" to "96%"]

Comments 17:[   It is also recommended that the abbreviations be revised and standardized, indicating the full term only the first time it appears in the manuscript and subsequently using the acronym.]

Response 17:[ Thank you very much for your comments. We have paid special attention to and improved the abbreviation of microplastics and nano-selenium.]

Comments 18:[The names of the genes should be updated to conform to the international nomenclature guidelines.]

Response 18:[ Thanks for your suggestion, we have modified the gene names according to the International Gene Naming Guidelines]

Reviewer 4 Report

Comments and Suggestions for Authors

The Authors did not correct or respond to my comments in any way. They even ignored the mistake in the name L54 - Betserius. After my comment, they confirmed that it refers to Jöns Jacob Berzelius. However, they did not even check the spelling. The results from the WB still raise significant doubts. I do not understand the translation related to the quality of the antibodies. If they are intended for WB and the procedure is performed correctly, the band should be visible at the appropriate height. The attached original images are bands cut out without ladders, so there is no way to verify if the expected protein is present.

Author Response

Comments 1:[The Authors did not correct or respond to my comments in any way. They even ignored the mistake in the name L54 - Betserius. After my comment, they confirmed that it refers to Jöns Jacob Berzelius. However, they did not even check the spelling. The results from the WB still raise significant doubts. I do not understand the translation related to the quality of the antibodies. If they are intended for WB and the procedure is performed correctly, the band should be visible at the appropriate height. The attached original images are bands cut out without ladders, so there is no way to verify if the expected protein is present.]

Response 1:[We greatly appreciate your professional comments on our articles. Based on your suggestions, we have made extensive revisions to the previous manuscript. But there may still be some carelessness and inadequacies, and we apologize again. Regarding the question of "Jons Jacob Berzelius", we have repeatedly checked the contents of the references and confirmed their consistency. But we are ashamed that we did not consider and recognize the misspelling of a person's name! We have now changed the name. Thank you for your scientific rigor in questioning the protein results. We performed Western blot experiments according to the correct steps. However, different proteins require different reaction times, and other situations cannot guarantee experimental consistency, so the original image may have some clear and some fuzzy situations, but we can guarantee that what we upload is the real original image. And we apologize for the original images are bands cut out without ladders. In order to ensure that the proteins with similar Kda react at the same temperature for the same time, we will cut the membrane in advance and then transfer the membrane. The reason for this is to save money and to ensure the consistency of experimental conditions. I hope you can understand. Finally, we sincerely thank you for your constructive comments on the manuscript, which will greatly help us to improve the quality of the manuscript!]

Round 3

Reviewer 4 Report

Comments and Suggestions for Authors

Dear Authors,

Unfortunately, I don't understand. The research group I belong to performs Western Blots, and this explanation does not convince me at all. It fails to address the concerns that were raised. I stand by my previous opinion.

with best regards